# Variations of VEGFR2 Chemical Space: Stimulator and Inhibitory Peptides

**DOI:** 10.3390/ijms25147787

**Published:** 2024-07-16

**Authors:** Claudiu N. Lungu, Ionel I. Mangalagiu, Gabriela Gurau, Mihaela Cezarina Mehedinti

**Affiliations:** 1Department of Functional and Morphological Science, Faculty of Medicine and Pharmacy, Dunarea de Jos University, 800010 Galati, Romania; gabriela.gurau@ugal.ro (G.G.); mihaela_hincu10@yahoo.com (M.C.M.); 2Faculty of Chemistry, Alexandru Ioan Cuza University of Iasi, 11 Carol 1st Bvd, 700506 Iasi, Romania

**Keywords:** peripheral artery disease, molecular modeling, docking, angiogenesis, vascular morphogenesis, chimeric protein

## Abstract

The kinase pathway plays a crucial role in blood vessel function. Particular attention is paid to VEGFR type 2 angiogenesis and vascular morphogenesis as the tyrosine kinase pathway is preferentially activated. In silico studies were performed on several peptides that affect VEGFR2 in both stimulating and inhibitory ways. This investigation aims to examine the molecular properties of VEGFR2, a molecule primarily involved in the processes of vasculogenesis and angiogenesis. These relationships were defined by the interactions between Vascular Endothelial Growth Factor receptor 2 (VEGFR2) and the structural features of the systems. The chemical space of the inhibitory peptides and stimulators was described using topological and energetic properties. Furthermore, chimeric models of stimulating and inhibitory proteins (for VEGFR2) were computed using the protein system structures. The interaction between the chimeric proteins and VEGFR was computed. The chemical space was further characterized using complex manifolds and high-dimensional data visualization. The results show that a slightly similar chemical area is shared by VEGFR2 and stimulating and inhibitory proteins. On the other hand, the stimulator peptides and the inhibitors have distinct chemical spaces.

## 1. Introduction

A well-coordinated response from cells is crucial for forming new blood vessels. This response involves specific receptors on the cell surface. Although the main molecular signals are known, their interaction mechanism is still not fully understood [1].

VEGF-VEGFR, Notch–DSL, Tie–Angiopoietin, VE–cadherin, and Ephrin–Eph are the main pathways in vascular morphogenesis. Among these, the VEGF pathway is essential for regulating angiogenesis, which consists of the growth of new blood vessels. It works by activating processes in endothelial cells that promote their growth, movement, and survival, as well as controlling vessel permeability [2].

VEGF mainly affects endothelial cells and influences other cell types like monocytes and macrophages. It promotes the growth and movement of endothelial cells in laboratory settings. The VEGF family includes five molecules: VEGF A to E. Each one binds to specific receptors on the cell surface, triggering a process that activates them through phosphorylation [3].

VEGF plays a crucial role in forming blood vessels by interacting with specific receptors on cell surfaces, leading to various cellular responses that promote angiogenesis [4,5].

VEGF, as stated, interacts with specific receptors on endothelial cells, mainly VEGFR2, to trigger cellular responses involved in angiogenesis. Although VEGFR1 signaling is less potent, it contributes to endothelial cell proliferation by merging with the VEGFR2 pathway. Activation of VEGFR2 leads to various downstream pathways that regulate cell survival, proliferation, and permeability. One pathway involves PI3K-AKT-mTOR signaling, while another crucial pathway involves the PLCI-mediated activation of PKC, leading to the induction of the ERK pathway. Endothelial cell migration is influenced by VEGFA/VEGFR2 signaling through p38MAPK activation. This signaling network plays a critical role in angiogenesis by regulating various enzymes, receptors, and transcription factors. Despite efforts, clinical success in promoting angiogenesis in peripheral artery disease patients remains challenging [6,7,8].

VEGF-A is vital for endothelial cell functions related to angiogenesis, primarily through VEGFA/VEGFR2 signaling, which drives endothelial cell proliferation, migration, survival, and new vessel formation. Cell signaling is tightly regulated spatially and temporally, with specialized membranes and vesicles containing specific lipids and proteins modulating signaling output. Phosphatidylinositol 4,5-bisphosphate (P.I. (4,5)P2) is crucial for multiple cellular processes. Furthermore, the tiny G proteins Rap1a and Rap1b offer insights into VEGF signaling in endothelial cells by playing critical roles in angiogenesis and endothelial cell responses to VEGF [9,10].

Furthermore, vascular endothelial cells have the glycoprotein VEGFR-2, which binds to VEGF-A. It is essential to angiogenesis and has particular autophosphorylation sites upon binding to VEGF. Compared to VEGFR1, an impaired tyrosine kinase receptor, VEGFR-2 is much more active. VEGF activates VEGFR2 on the membrane of endothelial cells, which starts a chain reaction of signaling molecules that include VRASP, PLCγ, ScK, Cdc42, Src, and PI3K. These molecules regulate cell migration, proliferation, survival, and permeability through interactions with downstream pathways such as ERK, p38MAPK, and AktPKE. Essential for both healthy and pathological angiogenesis, VEGFR-2 mediates VEGF-driven responses in endothelial cells [11,12].

In addition to VEGFR-2, the Notch signaling pathway plays a significant role in embryonic development. Delta Notch or Seratt-like ligands stimulate the Notch receptor, leading to downstream effects on DNA transcription factors like Hes1/5 and Hey. This pathway is essential for proper embryonic development [13].

VEGF-A, acting through VEGFR2, leads to endothelial cell proliferation, migration, survival, and new vessel formation, crucial for angiogenesis. Cell signaling in angiogenesis is tightly regulated and involves various molecules, including phosphatidylinositol 4,5-bisphosphate (P.I. (4,5)P2) and small G proteins like Rap1a and Rap1b. Hypoxia and downstream signaling pathways influence angiogenesis, including SOX17- and VEGF-R2-mediated pathways [14].

Furthermore, focal adhesion kinase (FAK) plays a crucial role in embryonic angiogenesis, regulating endothelial cell survival and barrier functions. Loss of FAK or its kinase activity decreases endothelial cell proliferation and migration, indicating FAK’s role as a kinase in regulating adult angiogenesis [15].

VEGFR-2 and other signaling pathways are essential targets for therapeutic strategies that promote angiogenesis and treat vascular diseases.

In this respect, ischemic diseases like heart failure, strokes, and peripheral artery disease result from poor blood supply. Treating these conditions with pro-angiogenic molecules is appealing. VEGFA plays a crucial role in vessel formation, growth, and branching, making it a critical pro-angiogenic molecule. It primarily acts on VEGFR2 but also stimulates VEGFR1. Targeting both receptors could be a promising therapy for promoting angiogenesis. Despite promising experimental results, there are currently no FDA-approved pro-angiogenic molecules. 

Extensive research has categorized various pro-angiogenic molecules, including angiogenic proteins, gene therapy, peptide drugs, and organic molecules [16,17].

Angiogenic proteins are essential molecules involved in angiogenesis, forming new blood vessels from existing ones. This mechanism is crucial for various physiological and pathological situations, including wound healing, embryonic development, and tumor growth. Examples of such proteins are Vascular Endothelial Growth Factor (VEGF), Basic Fibroblast Growth Factor (bFGF), Angiopoietins (Ang-1 and Ang-2), Platelet-Derived Growth Factor (PDGF), Transforming Growth Factor-Beta (TGF-β), Epidermal Growth Factor (EGF), Hepatocyte Growth Factor (HGF), and Matrix Metalloproteinases (MMPs). Regarding their structure, VEGF is a dimeric glycoprotein, often consisting of two identical monomers linked by disulfide bonds. The primary isoforms (e.g., VEGF-A) have variations in their sequence due to alternative splicing.

bFGF is a single-chain polypeptide with a compact, globular structure. It has a high affinity for heparan sulfate proteoglycans, which stabilize it and enhance its signaling. Angiopoietins are secreted glycoproteins with a characteristic structure, including an N-terminal superclustering domain, a central coiled-coil domain for dimerization, and a C-terminal Fibrinogen-like domain for receptor binding. PDGF is a dimeric protein consisting of A and B chains that can form homo- or heterodimers (e.g., PDGF-AA, PDGF-BB, PDGF-AB). Disulfide bonds link the monomers. TGF-β is a dimeric peptide, with each monomer having a cystine knot motif, a common feature among the TGF-β superfamily. The dimer is stabilized by disulfide bonds between the monomers. EGF is a small polypeptide consisting of 53 amino acids with three intramolecular disulfide bonds that create a compact, stable structure. The presence of these disulfide bonds is critical for its biological activity. HGF is a large, heterodimeric protein composed of an alpha chain (69 kDa) and a beta chain (34 kDa) linked by a single disulfide bond. The alpha chain contains four kringle domains and an N-terminal hairpin domain, while the beta chain has serine protease homology. MMPs are a family of zinc-dependent endopeptidases. They have a common structure consisting of a pro-domain (which maintains them in an inactive form), a catalytic domain with a zinc-binding motif, a hinge region, and a hemopexin-like C-terminal domain that contributes to substrate specificity and interaction with tissue inhibitors [18,19,20,21,22,23,24,25,26].

Regarding gene therapy, VEGF gene therapy, FGF gene therapy, HGF gene therapy, Angiopoietin gene therapy, PDGF gene therapy, and combined gene therapy are optimal angiogenesis solutions at various stages of research and clinical development, with some having shown promising results in early-phase clinical trials [27,28].

The most studied form of VEGF gene therapy involves the delivery of the VEGF-A gene to promote the formation of new blood vessels. Clinical trials have tested its efficacy in treating peripheral arterial disease and myocardial ischemia. VEGF-C and VEGF-D are targeted for lymphangiogenesis and angiogenesis. These genes have been explored for treating lymphedema and enhancing wound healing. FGF-1 (Acidic Fibroblast Growth Factor) gene therapy delivers the FGF-1 gene, aims to enhance angiogenesis, and has been tested in clinical trials for treating coronary and peripheral artery disease. Gene therapy using FGF-2 has shown potential in promoting angiogenesis and improving blood flow in ischemic tissues [24,29]. HGF gene therapy is designed to stimulate angiogenesis and has been evaluated in clinical trials for its potential to treat ischemic heart disease and peripheral artery disease [30,31].

Angiopoietin-1 (Ang-1) therapy stabilizes newly formed blood vessels and promotes vascular maturation. It has been studied for its potential to enhance angiogenesis in ischemic tissues and improve tissue repair [32]. Platelet-derived growth factor-B (PDGF-B) therapy involving PDGF-B has been explored for its role in recruiting pericytes and smooth muscle cells, stabilizing new blood vessels, and improving blood flow in ischemic tissues [33,34,35,36]. Hypoxia-Inducible Factor-1 Alpha (HIF-1α) gene therapy involves the delivery of HIF-1α, a transcription factor that induces the expression of several angiogenic factors, including VEGF, under hypoxic conditions. It has been studied for treating ischemic cardiovascular diseases. Some therapies synergize VEGF and FGF genes to promote angiogenesis and enhance therapeutic outcomes [37,38]. Combining these genes encourages the formation and stabilization of new blood vessels, providing a more robust angiogenic response [39,40].

Pro-angiogenic peptide drugs are designed to promote the formation of new blood vessels and have potential applications in treating various conditions such as ischemic diseases, wound healing, and tissue regeneration. Thymosin Beta-4 is a small, 43-amino-acid peptide. TB-4 promotes angiogenesis by enhancing endothelial cell migration and differentiation. It also regulates actin polymerization and promotes wound healing [41]. VEGF mimetic peptides are designed to mimic the active site of VEGF and typically consist of short sequences derived from the VEGF protein. VEGF mimetic peptides bind to VEGF receptors, activating them to stimulate angiogenesis and endothelial cell proliferation [24]. Angiopoietin-derived peptides are derived from Angiopoietins, particularly the receptor-binding regions of Ang-1 or Ang-2. They mimic the action of Angiopoietins, promoting blood vessel maturation and stability. Ang-1-derived peptides are especially noted for enhancing vascular stabilization [42]. Hepatocyte Growth Factor (HGF) mimetic peptides are short peptides derived from the active regions of HGF. HGF mimetic peptides activate the c-Met receptor, promoting angiogenesis and enhancing tissue repair and regeneration [43]. Fibroblast Growth Factor (FGF)-derived peptides are derived from FGF, particularly the regions that interact with FGF receptors. FGF-derived peptides stimulate endothelial cell proliferation and differentiation, promoting angiogenesis and tissue repair [44]. R-spondin peptides are a family of secreted proteins, and peptides derived from them are designed to activate the Wnt signaling pathway. R-spondin peptides promote angiogenesis through the activation of Wnt signaling, which is involved in endothelial cell proliferation and migration [45].

Pro-angiogenic organic molecules, often small molecules, are designed to promote angiogenesis through various mechanisms. Thalidomide has a glutarimide ring attached to a phthalimide ring. Initially known for its teratogenic effects, thalidomide has been found to promote angiogenesis under certain conditions by increasing the expression of VEGF and other pro-angiogenic factors [46]. Vandetanib is a quinazoline derivative. Vandetanib is a tyrosine kinase inhibitor that targets VEGFR, EGFR, and RET kinase, promoting angiogenesis by upregulating VEGF signaling pathways [47]. Sorafenib is a biaryl urea. Sorafenib inhibits multiple kinases involved in angiogenesis, including VEGFR and PDGFR. This inhibition can paradoxically lead to pro-angiogenic effects in certain contexts, such as by promoting a more normalized vascular environment [48]. Lenalidomide is a derivative of thalidomide with an isoindolinone structure. Lenalidomide enhances angiogenesis by increasing VEGF production and other growth factors, similar to thalidomide but with improved safety and efficacy profiles [49]. Bevacizumab is a monoclonal antibody. Although primarily an anti-angiogenic agent targeting VEGF-A, in certain dosages and contexts, it can paradoxically promote angiogenesis by modifying VEGF signaling and vascular normalization [50]. 2-Methoxyestradiol (2-ME2) is an endogenous estrogen metabolite. 2-ME2 promotes angiogenesis by stabilizing HIF-1α and upregulating VEGF. It also modulates microtubule dynamics [51].

Peptides, smaller molecules than proteins, do not require complex structures to be biologically active. They can be manipulated easily and optimized to mimic angiogenesis-stimulating molecules. Peptides can also be modified or conjugated with other molecules to enhance their properties. Due to their simplicity and smaller size, pro-angiogenic peptides can be rapidly synthesized to stimulate angiogenesis effectively [52].

This computational study aims to characterize the chemical space of stimulant and inhibitory VEGFR2 proteins to further design a potent peptide or organic molecule that can shape the angiogenesis and vascularmorphogenesis processes.

## 2. Results

As presented in the Section 4, a complete series of 3D protein molecule structures (PDB) that act on VEGF2 and consecutively inhibit or stimulate vascular morphogenesis have been used. Homology modeling was used for some structures to generate 3D molecules using their Uniprot ID (where the PDB structure was unavailable). As stated, homology modeling for the angiogenesis inhibitor Vasstatin and angiogenesis stimulators PDGFC, PIGF, and PDGF D was performed using their Uniprot IDs. The resulting structures are shown below (Figure 1):

To explore the VEGFR2 interaction with organic molecules, the VEGFR2-binding site was determined computationally to perform docking studies. Furthermore, the binding site characterization (VEGFR2 as the target molecule) retrieved the following binding sites for the VEGFR2 PDB model 3VNT [53]: (a) a major cavity 1 with a volume =435.712 and the following coordinates x = 29.14, y = −36.14, z = −18.65; (b) cavity 2 with a volume = 23.04, x = 23.25. z = −44.35, y = −14.55; (c) cavity 3 with volume = 16.384, x = 11.72, y = −32.57, z = −21.62; (d) cavity 4 with a volume =5.36, x = 13.22, y = −12.71, z = −18.12; (e) cavity 5 with a volume =10.24, x = 12.46, y = −7.67, z = −29.09; (f) cavity 6 with a volume = 12.80 with x = 26.74, y = −47.08, z = −27.29. Cavity 1 was chosen for docking studies, taking into account its volume (Figure 2).

Docking results of 27/520 (see the rest of docking results in Appendix A) structures selected randomly using the Chembl_1 database are shown in Figure 3: 

The protein–protein docking results are shown in Figure 3 and Figure 4. Also, in Figure 4, a protein–protein complex is displayed as an example of the docking of inhibitory and stimulant proteins. 

Protein–protein docking for the inhibitory protein structures against VEGFR2 retrieved the following complexes with the binding energies represented in Figure 5:

Protein–protein docking for the inhibitory protein structures against VEGFR2 retrieved the following complexes with the binding energies represented in Figure 6:

Furthermore, the chimeric homology model computed for the inhibitory protein structures using their Aa sequences has a sequence similarity of 99.99%, a molecular probability score of 2.03, a general clash score of 3.99, a Ramachandran plot favored score of 93.76%, a rotamer outlier of 3.75%, an aC beta deviation of 2, a ratio of bad bounds to favorable bounds of 2/3449, and a ratio of bad angles to favorable angles of 20/4662 (results obtained using mol Probability version 4.1, as stated in the Section 4). As stated in the Section 4, the chimeric model was further optimized using the Swiss online preparation server. The structure is represented in Figure 7a.

The chimeric homology model design for the stimulatory protein structure using their Aa sequences has a sequence similarity of 99.99%, a mol probability score of 1.84, a general clash score of 1.77, a Ramachandran favored score of 94.42%, a rotamer outlier of 1.40%, an aC beta deviation of 5, a ratio of bad bounds to favorable bounds of 0/1810, and a ratio of bad angles to favorable angles of 11/2411 (results obtained using mol Probability version 4.1). The chimeric model was further optimized using the online preparation server Swiss Model. The structure is represented in Figure 7b.

Furthermore, Table 1 shows the Aa sequences of the inhibitory and stimulatory chimeric models. Figure 8 and Table 2 show the Aa composition and properties of the chimeric models. 

Docking results of the inhibitory and stimulatory model show the following: the complex between the inhibitory model and VEGFr2 has a total energy of −71.62 kcal/mol, and the complex of the stimulatory chimeric model and VEGFR2 has a total energy of −58.81 kcal/mol.

Furthermore, the chemical space characterized by molecular descriptors for angiogenesis inhibitor molecules and angiogenesis stimulator molecules, respectively, is represented in Figure 9.

Figure 10 represents the chemical space of the chimeric inhibitory and stimulant models. 

Furthermore, the C-alpha-based distance plot computed for the chimeric inhibitory and stimulant models and plots for PEDF and Angiopoietin 1 are represented in Figure 11. 

In Figure 12, the chimeric models’ multidimensional data are represented.

The polynomial equations resulting from the inhibitory, stimulant, and combined multidimensional spaces are shown below: **Inhibitory space** y = −23.758 × 6 − 3.4701 × 5 + 12.001 × 4 − 0.9262 × 3 − 2.8557 × 2 + 0.4032 × + 0.1676 (1)
**Stimulant space** y = −1.1017 × 6 − 3.6244 × 5 + 2.7119 × 4 + 0.7384 × 3 − 1.2141 × 2 − 0.269 × + 0.0601 (2)
**Combine space** y = −7.9346 × 6 − 9.1068 × 5 + 6.8296 × 4 + 1.3786 × 3 − 2.1349 × 2 + 0.0735 × + 0.1191 (3)

Also, a map of the 2D complex space is shown in Figure 13 below: 

## 3. Discussion

In structural biology, homology modeling, sometimes called comparative modeling, is a computational technique that predicts a protein’s three-dimensional structure using its amino acid sequence and the structure of a comparable protein known to exist (template). The fundamental premise is that proteins with similar sequences frequently exhibit structural and functional similarities. The following steps are usually involved in the homology modeling process. Finding a template entails finding an appropriate homologous template—comparable in sequence and structure—to the target protein that has a known three-dimensional structure. Numerous databases and sequence alignment techniques, such as BLAST (Basic Local Alignment Search Tool) and HHpred (Homology Detection and Structure Prediction by HMM-HMM Comparison), can be used. When the target protein’s amino acid sequence matches the template protein, this is known as sequence alignment. This alignment is essential to map the template structure onto the target protein. With model building based on sequence alignment, a three-dimensional model of the target protein is constructed using computational techniques such as comparative modeling algorithms. These algorithms use the known structure of the template protein to generate a model of the target protein by aligning corresponding residues and building missing regions. Model refinement is where the initial model may undergo refinement to improve its quality and accuracy. This can involve energy minimization, molecular dynamics simulations, and other optimization techniques to optimize the geometry and remove steric clashes. Lastly, the quality of the homology model is assessed using various validation criteria such as Ramachandran plot analysis, MolProbity scores, and QMEAN scores. These measures help evaluate the stereochemical quality and overall reliability of the model. The validation of homology modeling involves assessing the quality and reliability of the predicted protein structure. Several techniques and criteria can be used: (a) Ramachandran plot analysis evaluates the amino acid residues’ backbone dihedral angles (φ and ψ) in the modeled structure. The Ramachandran plot shows allowed and disallowed regions based on stereochemical constraints. A high percentage of residues in the favored areas indicates a good-quality model. (b) MolProbity assesses the overall quality of protein structures, including homology models, by evaluating steric clashes, bond lengths, bond angles, and other geometric parameters.

Lower MolProbity scores indicate better model quality. (c) QMEAN (Qualitative Model Energy ANalysis) is a composite scoring function that evaluates the overall model quality based on various structural features, including energy terms, solvation, and torsion angles. Higher QMEAN scores correspond to better-quality models. (d) ProSA-web: ProSA-web calculates the Z-score of the modeled structure, which measures its overall energy deviation from experimental structures of similar size. Lower Z-scores indicate better agreement with experimental structures. This study used Ramachandran plots to validate the homology models [54]. The Ramachandran plots are represented in Figure 14.

As observed in Figure 14, a high percentage of residues in the favored regions indicates a good-quality model. Also, the homology models obtained are stable and have an energetically favorable profile. 

Binding cavities often have unique structural features, allowing them to interact with specific molecules. These features include pockets, grooves, and specific amino acid residues that form hydrogen bonds, hydrophobic interactions, or electrostatic interactions with the ligand. Binding cavities exhibit specificity towards particular ligands. This specificity arises from complementary shapes and chemical properties between the cavity and the ligand. The binding of ligands to these cavities often triggers conformational changes in the protein, leading to its activation or inhibition. This functional modulation is crucial for various biological processes, including enzymatic reactions, signal transduction, and molecular transport. Binding cavities are frequently targeted by drugs and therapeutics to modulate protein function. Small molecules or medicines can be designed to bind to these cavities, either activating or inhibiting the protein’s activity. Binding cavities may exhibit flexibility or adaptability to accommodate different ligands or undergo conformational changes upon ligand binding. This flexibility is essential for the protein to perform its biological functions effectively. In addition to the primary binding site, proteins may possess allosteric sites distinct from the active site. However, they can regulate the protein’s activity through conformational changes induced by ligand binding at these sites [55].

Furthermore, protein–protein interactions (PPIs) are fundamental in virtually all biological processes, including cell signaling, gene regulation, enzymatic activity, and structural support. These interactions occur when two or more proteins bind together transiently or stably to form complexes, enabling them to carry out specific functions within the cell. Understanding protein–protein interactions is crucial for elucidating cellular processes and designing therapeutics to modulate these interactions for various purposes. PPIs can be classified into several types based on duration, strength, and functional consequences. These include transient interactions, such as signaling interactions, and stable interactions, such as those involved in forming structural complexes. Protein–protein interactions typically occur through specific binding interfaces, where complementary surfaces of the interacting proteins come into contact. These interfaces often involve amino acid residues that form hydrogen bonds, hydrophobic interactions, electrostatic interactions, or van der Waals forces. PPIs exhibit specificity, meaning that proteins selectively interact with their binding partners. This specificity arises from complementary shapes, charges, and chemical properties between the interacting proteins. The interactions between proteins can be regulated dynamically in response to various cellular signals, environmental cues, or post-translational modifications. This regulation allows cells to fine-tune their signaling pathways and responses to internal and external stimuli. Protein–protein interactions mediate various biological processes, including enzyme activation/inhibition, signal transduction, protein trafficking, DNA replication and repair, and cytoskeletal organization. Disruption or dysregulation of these interactions can lead to diseases such as cancer, neurodegenerative disorders, and autoimmune diseases [56].

Several residues on VEGFR2 have been identified as involved in protein–protein interactions (PPIs), particularly with its ligands (Vascular Endothelial Growth Factors, VEGFs) and other signaling molecules. While the specific residues involved may vary depending on the interaction partner and context, here are some general insights into the regions and residues of VEGFR2 involved in PPIs. The extracellular domain of VEGFR2 interacts with VEGF ligands, typically homodimers or heterodimers. Specific residues within the extracellular domain of VEGFR2 bind to VEGF. For example, residues in the ligand-binding domain (LBD), including those in Ig-like domains, have been implicated in VEGF binding. The intracellular tyrosine kinase domain of VEGFR2 is involved in downstream signaling cascades following ligand binding. This domain can interact with various signaling proteins through phosphorylation-dependent or -independent interactions, including adaptor molecules and other kinases. Specific residues within the TKD may participate in these interactions, particularly those in substrate recognition and catalysis. VEGFR2 undergoes autophosphorylation on specific tyrosine residues within its intracellular domain upon ligand binding. These phosphorylated tyrosine residues serve as docking sites for downstream signaling proteins containing SH2 (Src homology 2) or PTB (phosphotyrosine-binding) domains, mediating protein–protein interactions critical for signal transduction. Through direct or indirect interactions, VEGFR2 can form complexes with other receptors or co-receptors, such as neuropilins, integrins, and other RTKs. Adaptor proteins or scaffolding molecules often mediate these interactions, and specific residues within VEGFR2 may contribute to the stability or specificity of these complexes. VEGFR2 contains regulatory domains, such as the juxtamembrane and kinase insert domains, which may participate in protein–protein interactions that modulate the receptor’s activity, localization, or stability. In this study, 12 protein–protein docking studies were performed on inhibitory protein complexes and 14 on stimulant protein complexes. All the protein docking studies retrieved stable VEFFR2–protein complexes. 

In Figure 5 and Figure 6, the protein–protein docking results are displayed. In Figure 5, VEGFR2 docked with the inhibitory proteins is shown. The best docking energies (kcal/mol) are observed when VEGFR2 is docked with 1AU1, and the most considerable complex energy is observed at the VEGFR2-1BBN complex. However, all complexes display favorable energies with presumably notable biological activity. In Figure 6, VEGFR2 is docked with the stimulant proteins. The protein 2X1W forms the most favorable complex with VEGFR2. In this case, 2TGP forms the lowest-energy complex (kcal/mol). Like in the case of inhibitory proteins, all complexes are energetically favorable. If a complex has a negative total energy, it generally indicates that the interactions within the complex are favorable and that the complex is stable. Negative total energy suggests that the attractive forces (such as electrostatic interactions, hydrogen bonding, and van der Waals interactions) between the molecules in the complex outweigh the repulsive forces (such as steric hindrance or electrostatic repulsion). These favorable interactions contribute to the stability of the complex. A negative total energy often correlates with a strong binding affinity between the molecules in the complex. The stronger the binding affinity, the more negative the total energy tends to be. This indicates that the complex will likely form and persist under given conditions. In thermodynamic terms, a negative total energy corresponds to a decrease in the overall free energy of the system upon complex formation. This suggests that the complex is stable under the prevailing conditions and that the formation of the complex is thermodynamically favorable. It is important to note that the accuracy of energy calculations depends on the methods used for computation (e.g., quantum mechanical calculations, molecular mechanics simulations). Different computational methods may yield different absolute energy values, but the relative energy values (such as the change in energy upon complex formation) are generally more meaningful. While a negative total energy indicates stability, it does not necessarily guarantee biological activity or function. Experimental validation is often required to confirm the biological relevance of a predicted complex. Additionally, factors such as entropy and solvent effects, which are not always fully accounted for in energy calculations, can influence the stability of complexes in biological systems. Solvation energy refers to the energy change associated with the process of solvation, where solvent molecules surround and interact with solute molecules to form a solution. It plays a crucial role in various chemical and biochemical processes, influencing the stability, solubility, and reactivity of solutes in solution. Solvation energy can be either favorable (exothermic) or unfavorable (endothermic) depending on the nature of the solute–solvent interactions. Solvation energy is the difference in energy between the solvated and separated states of solute and solvent molecules. It represents the overall effect of solvent molecules stabilizing or destabilizing the solute. When solvent molecules interact favorably with the solute, solvation energy is negative (exothermic), indicating that the solvated state is more stable than the separated state. This typically occurs when solute–solvent interactions are strong, such as in the case of polar solutes dissolving in polar solvents or nonpolar solutes dissolving in nonpolar solvents. Conversely, when solvent–solute interactions are weak or repulsive, solvation energy is positive (endothermic), indicating that the solvated state is less stable than the separated state. This may occur when dissolving nonpolar solutes in polar solvents or polar solutes in nonpolar solvents, where the interactions between unlike molecules are less favorable. The magnitude of solvation energy depends on various factors, including the nature of solute and solvent molecules, their polarity, size, shape, and temperature and pressure conditions. Solvation energy influences the rates and equilibrium of chemical reactions occurring in solution. Solvation of reactant molecules can either enhance or hinder their reactivity by stabilizing or destabilizing their transition states and intermediate species. In summary, angle energy is the potential energy associated with deviations of bond angles from their equilibrium values within a molecule. It is an important component of the total potential energy in molecular mechanics simulations and is crucial in determining molecules’ conformational stability and behavior. The specific form of the angle energy term varies depending on the force field being used. However, in general, it represents the energy associated with the bending or stretching of bonds and contributes to the overall potential energy of the molecular system. In a molecular system, chemical bonds connect atoms, and these bonds have characteristic bond angles. The angle energy arises from the deviation of these bond angles from their preferred or equilibrium values. When the bond angles deviate, the system’s potential energy increases, contributing to the overall energy of the molecule. Here, both angular and solubility energies show favorable values that correlate with the total complex energies. Overall, docking results show that the docking procedure was performed properly. Finally, VEGFR2 forms stable active complexes with the inhibitory and stimulant peptides retrieved from the literature [57,58,59]. However, all complexes of the inhibitory and stimulatory proteins display favorable energies with presumably notable biological activity. Regarding inhibitory molecule docking energies, the most favorable energy is observed at 4EB1 with a total complex energy of −92.87 kcal/mol. The highest docking energy at a stimulant molecule is observed for 2X1W with a docking energy of −99.99 kcal/mol. Also, in the case of inhibitors, the most favorable solvation energy is observed at 4EB1 with 15,734.68 kcal/mol. The same is true in the case of the stimulants; the most favorable docking energy is observed at 2XIW with −14,554.78 kcal/mol.

Docking studies have certain drawbacks, including imprecise scoring functions, insufficient consideration of protein flexibility and solvent effects, and restricted conformational sampling, all of which can result in inaccurate predictions. Docking studies frequently offer a fixed image and fail to consider the dynamic and intricate characteristics of biological interactions. These constraints can affect the dependability of the outcomes [60,61].

However, the utilization of docking in drug design is restricted to biological targets that have known crystal structures. Various methods have been employed to address this specific constraint. One way to overcome the lack of 3D structures is to create homology models using structural templates that have very similar sequences. In addition, these techniques can be employed in conjunction with molecular dynamics (MD) to corroborate and enhance the accuracy of the computationally simulated complexes [62,63]. However, the current advancements in structural biology and crystal structure determination, which are steadily improving the availability of experimentally obtained ligand–target complexes, will undoubtedly alleviate this problem. Computational techniques, such as molecular dynamics, have been extensively employed to investigate the conformational space of the targets, ligands, and ligand–target complexes. This allows for a more accurate description of the dynamic behavior of ligand–target complexes and improves the precision of docking results [64,65]. In this respect, the computational studies in this work have been performed using crystallographic models with the best resolution possible. Also, the protein–protein complexes were selected based on the most favorable complex energies. Furthermore, each complex was subject again to energy minimization and structural error detection methodologies 

In a chimeric model, structural elements from different molecules are combined to create a new molecule with desired characteristics. This could involve combining functional groups, binding pockets, or other molecular features from existing molecules to generate a hybrid structure. Chimeric models are often designed based on a rational understanding of molecular interactions and structure–activity relationships. Researchers may select specific elements from different molecules known to interact with a target protein or exhibit certain biological activities. Chimeric models can be subjected to virtual screening techniques to assess their potential for binding to a target protein or modulating a biological pathway. Computational methods such as molecular docking or molecular dynamics simulations can be employed to predict the binding affinity and mode of interaction of the chimeric molecule with its target. Chimeric models are valuable tools in drug design and discovery. By combining elements from different molecules, researchers can create novel compounds with improved potency, selectivity, or pharmacokinetic properties compared to existing drugs. Chimeric models can be used in lead optimization, where initial hits identified through high-throughput screening are modified to enhance their drug-like properties. Chimeric molecules may undergo iterative rounds of computational design, synthesis, and biological testing to optimize their activity and pharmacological profile [66,67,68].

Comparing two amino acid (Aa) sequences is fundamental in bioinformatics and molecular biology. Sequence comparison allows researchers to identify similarities, differences, and patterns between proteins, which can provide insights into their structure, function, and evolutionary relationships. Two Aa sequences can be compared as follows: Perform a pairwise alignment of the two Aa sequences using algorithms such as Needleman–Wunsch, Smith–Waterman, or FASTA. These algorithms identify the optimal alignment between the sequences by maximizing the number of matched residues and minimizing gaps and mismatches.

Use scoring matrices such as BLOSUM or PAM to assign scores to matches, mismatches, and gap penalties during sequence alignment. These matrices are based on empirical observations of amino acid substitutions in related proteins and help quantify the similarity between sequences. Calculate sequence similarity and identity scores based on the alignment results. Sequence similarity is the percentage of identical residues and conservative substitutions between the sequences, while sequence identity is the percentage of identical residues only. Similarity and identity scores provide quantitative measures of the degree of similarity between sequences and can help compare proteins with different evolutionary distances. Identify functional domains, motifs, and conserved regions within the aligned sequences. Conserved areas often correspond to functional domains or motifs essential for protein structure and function. Use tools like InterPro, Pfam, or SMART to annotate domains and motifs based on the alignment results. Perform phylogenetic analysis using the aligned sequences to infer evolutionary relationships between proteins. Phylogenetic trees can help elucidate protein sequences’ evolutionary history and divergence. Phylogenetic analysis can be conducted using software packages such as MEGA, PHYLIP, or RaxML [69,70].

The domain analysis of the Aa inhibitory chimeric model reveals that the representative domain is the serpin Ci1 domain. The serpin (serine protease inhibitor) family is a protein group that plays a crucial role in regulating proteolytic processes in various biological systems. Serpins are characterized by their ability to inhibit serine proteases, a class of enzymes involved in a wide range of physiological processes, including blood coagulation, immune response, inflammation, and tissue remodeling. Serpins typically share a conserved structure of around 350–400 amino acids. They fold into a compact, globular conformation with three β-sheets (A, B, C) and nine α-helices (A-I). The serpin fold contains a reactive center loop (RCL), which acts as bait for serine proteases. The RCL undergoes a conformational change upon protease binding, forming a covalent complex between the serpin and protease. Serpins inhibit serine proteases by a suicide substrate-like mechanism. Upon binding to the protease, the RCL of the serpin is cleaved by the protease, leading to the formation of an acyl–enzyme intermediate. This intermediate is then inserted into the central β-sheet of the serpin, irreversibly trapping and inactivating the protease. The serpin family is highly diverse and includes members with many functions beyond protease inhibition. Some serpins act as inhibitors of blood coagulation factors (e.g., antithrombin), while others regulate immune responses (e.g., α1-antitrypsin), inflammation, and tissue remodeling. Additionally, certain serpins have non-inhibitory functions, such as hormone transport (e.g., thyroxine-binding globulin) and chaperone-like activity. Mutations in serpin genes can lead to various diseases and disorders. For example, mutations in SERPINA1, encoding α1-antitrypsin, are associated with liver and lung diseases, including alpha-1 antitrypsin deficiency. Similarly, hereditary angioedema, a rare illness characterized by recurrent episodes of swelling in diverse body areas, can be brought on by mutations in SERPING1, the gene that codes for the C1 inhibitor. The serpin family has a long evolutionary history, and members can be found in various animals, including humans and microbes. Throughout their evolutionary history, serpins have undergone significant gene duplication, diversification, and specialization, giving rise to functionally unique subfamilies [71,72]. The antithrombin III domain is the domain of the serine protease inhibitor family. Thrombin, a crucial protease in the coagulation cascade, is inhibited by antithrombin III. Thrombin possesses non-hemostatic properties, such as regulating the behavior of endothelial cells, and is involved in the creation of blood clots. ATIII indirectly influences angiogenesis and endothelial cell function by blocking thrombin. It has been demonstrated that antithrombin III interacts with endothelial cells and modifies their activities. It can lessen endothelial cell proliferation, prevent leukocyte adherence to endothelial cells, and lessen endothelial cell production of growth factors and pro-inflammatory cytokines. These factors may impact vascular remodeling and angiogenesis. Because of its anti-inflammatory qualities, antithrombin III may indirectly affect angiogenesis. Angiogenesis and inflammation are intimately related, and vascular morphogenesis may be influenced by substances that reduce inflammation. The regulating function of ATIII in angiogenesis may be facilitated by its capacity to suppress inflammation. The significance of antithrombin III in preserving vascular homeostasis is underscored by the fact that dysregulation of its levels or function can result in thrombotic diseases or excessive bleeding. A higher risk of venous thromboembolism and other thrombotic problems is linked to antithrombin III deficiency. While antithrombin III’s role in vasculogenesis and angiogenesis is not as well studied compared to other angiogenic factors, emerging evidence suggests its involvement in modulating endothelial cell function and vascular remodeling processes. Further research is needed to elucidate the precise mechanisms through which ATIII influences vascular morphogenesis and its potential therapeutic implications for angiogenesis-related disorders. The domain analysis of the stimulant chimeric model suggests that the representative domain is Fibrinogen C2, the domain is Fibrinogen c, and the conserved sites are Fibrinogen. Fibrinogen, a glycoprotein found in blood plasma, plays a pivotal role in blood clotting (coagulation) by converting into fibrin during coagulation. Fibrinogen’s involvement in vascular morphogenesis, specifically in angiogenesis (forming new blood vessels from pre-existing ones), is less direct than its role in coagulation. However, emerging research suggests that Fibrinogen and its degradation products can influence angiogenesis through various mechanisms: Fibrinogen has been shown to exhibit pro-angiogenic properties. Studies have demonstrated that Fibrinogen-derived peptides can promote endothelial cell proliferation, migration, and tube formation, which are essential steps in angiogenesis. These peptides may act through specific receptors or signaling pathways on endothelial cells to stimulate angiogenesis [73,74,75]. During coagulation, Fibrinogen is converted into fibrin by the action of thrombin. The resulting fibrin forms a matrix, providing a scaffold for platelets and other blood components to adhere to and form a stable blood clot. This fibrin matrix provides a provisional matrix for endothelial cells to migrate and proliferate during angiogenesis. Fibrin degradation products, generated by the action of fibrinolytic enzymes such as plasmin, can modulate angiogenesis. These degradation products, including fibrin degradation products (FDPs) and fibrin-derived peptides, possess bioactive properties and can influence endothelial cell behavior, vascular permeability, and angiogenic signaling pathways. Fibrinogen and fibrin can interact with various growth factors, cytokines, and extracellular matrix components that regulate angiogenesis. Fibrinogen, for instance, can bind and alter the bioavailability of angiogenic molecules, including Fibroblast Growth Factor (FGF) and Vascular Endothelial Growth Factor (VEGF), which in turn affects angiogenic processes. Angiogenesis is necessary to provide oxygen and nutrients to the healing tissues, while fibrin and Fibrinogen play important roles in wound healing and tissue repair. To aid in tissue regeneration, the fibrin matrix that forms at the site of damage serves as a temporary scaffold for angiogenesis and encourages endothelial cell migration and proliferation. While Fibrinogen’s primary role is in blood clotting, its involvement in angiogenesis and vascular morphogenesis is increasingly recognized. Further research is needed to elucidate the precise mechanisms by which Fibrinogen and its degradation products influence angiogenesis and their potential therapeutic implications for angiogenesis-related disorders such as wound healing, cancer, and cardiovascular diseases. The resulting inhibitory chimeric model is larger than the stimulant chimeric model. 

In Figure 8, the Aa composition of both chimeric models is represented, showing that the inhibitory ceramic model has more Ala, Arg, Gly, Leu, Tyr, and Val than the stimulant chimeric model. For example, arginine and tyrosine residues are often involved in protein–protein interactions and molecular recognition processes, so a protein with more of these residues may have altered binding capabilities compared to a protein with fewer of these—amino acids such as glycine, alanine, and leucine influence protein structure. Glycine is highly flexible due to its small size, alanine is commonly found in protein helices, and leucine is frequently found in protein hydrophobic cores. Therefore, differences in the abundance of these amino acids could affect the structural characteristics of the proteins. 

The stimulant chimeric model has more Cys, Glu, Lys, Pro, Serr, Thr, and Trp. Cysteine residues are crucial for forming disulfide bonds in proteins, contributing to their structural stability and function. Proteins containing disulfide bonds play roles in angiogenesis by modulating growth factor signaling, extracellular matrix (ECM) assembly, and cell–matrix interactions [76]. Glutamate participates in various signaling pathways involved in cell proliferation, migration, and survival. Glutamate receptors and transporters expressed in endothelial cells regulate angiogenic responses by modulating intracellular calcium levels, nitric oxide (NO) production, and vascular permeability [77]. Lysine residues are abundant in extracellular matrix (ECM) proteins such as collagens, Fibrinogen, and fibronectin, which provide structural support for blood vessels. During angiogenesis, ECM proteins containing lysine residues regulate endothelial cell adhesion, migration, and tube formation [78]. Proline-rich motifs are found in angiogenic factors, cytokines, and extracellular matrix (ECM) proteins involved in vascular remodeling. Proline-rich proteins contribute to proteins’ structural stability and flexibility, including those involved in angiogenesis [79]. Serine and threonine residues are protein phosphorylation sites regulating angiogenic signaling pathways. Protein kinases and phosphatases that target serine/threonine residues modulate endothelial cell behavior, proliferation, and migration during angiogenesis [80]. Tryptophan metabolism and signaling pathways have been implicated in angiogenesis, inflammation, and immune responses. Tryptophan metabolites such as kynurenine and serotonin can regulate endothelial cell function, vascular permeability, and angiogenic responses [81]. 

Protein isoelectric point (pI) is crucial in drug design and formulation. For instance, in a study by Böttcher et al. (2010) [82], the authors designed peptides targeting the cell-penetrating peptide transporter, PepT1, by considering the pI of both the peptide and the transporter. By ensuring that the peptide had a different charge from PepT1 at physiological pH, they aimed to enhance peptide transport across cell membranes. This demonstrates how knowledge of pI can guide the design of molecules for improved drug delivery and efficacy. So, proteins’ isoelectric point (pI) is critical in various biological processes, including protein–protein interactions, enzyme–substrate interactions, and protein localization within cells. 

For example, in a study by Kyte and Doolittle (1982) [83], the authors investigated the role of pI in predicting transmembrane segments in proteins. They found that the distribution of charged residues relative to the pI could provide insights into the topology of membrane proteins, aiding in their prediction and understanding of membrane protein function [83]. A protein’s isoelectric point (pI) is the pH at which it carries no net electrical charge. Proteins with different pI values have different charge distributions at a given pH. If one protein has a pI of 7.0 and another has a pI of 8.3, presumably, the inhibitory chimeric model with a pI of 7.0 will have a zero net charge when the surrounding pH is adjusted to 7.0. At pH values below 7.0, the protein will carry a net positive charge due to more positively charged amino acids (e.g., lysine, arginine) than negatively charged ones (e.g., aspartic acid, glutamic acid).

Conversely, at pH values above 7.0, the protein will carry a net negative charge due to the dominance of negatively charged amino acids. Thus, at pH 7.0, the protein will be least soluble in water and may precipitate out of the solution. The stimulant chimeric model with a pI of 8.3 will carry no net charge at pH 8.3. At pH values below 8.3, the protein will take a net positive charge, while at pH values above 8.3, it will carry a net negative charge. Similarly to the protein with a pI of 7.0, at its pI (pH 8.3), the protein will be least soluble in water. Comparing these two proteins, the protein with a pI of 7.0 will have a net positive charge at physiological pH (around 7.4) and tend to interact more strongly with negatively charged molecules or surfaces. The protein with a pI of 8.3 will have a net negative charge at physiological pH and tend to interact more strongly with positively charged molecules or surfaces. Understanding the pI values of proteins is crucial for various applications, including protein purification, characterization, and predicting their behavior in different biological environments. It allows researchers to manipulate pH conditions to control proteins’ solubility, stability, and interactions in biochemical experiments and biotechnological applications [84].

The term “Total number of negatively charged residues (Asp + Glu)” refers to the sum of two specific amino acids: aspartic acid (Asp) and glutamic acid (Glu). These amino acids are considered negatively charged because they contain carboxyl groups that can ionize, releasing a hydrogen ion (H+) and resulting in a negatively charged carboxylate group (COO-). In proteins, aspartic acid and glutamic acid contribute to the protein molecule’s overall charge depending on the surrounding environment’s pH. These residues tend to be deprotonated at a pH above their respective pKa values (at which 50% of the molecules are deprotonated), carrying a negative charge. They tend to be protonated at a pH below their pKa values, carrying no net charge. A protein’s total number of negatively charged residues (Asp + Glu) is essential for understanding its overall charge distribution. It can influence various biological functions, interactions with other molecules, and the protein’s behavior under different pH conditions. Proteins with many negatively charged residues may interact preferentially with positively charged molecules or surfaces. In contrast, proteins with many positively charged residues may interact preferentially with negatively charged molecules or surfaces. In summary, the total number of negatively charged residues (Asp + Glu) provides valuable information about the charge distribution of a protein and its potential interactions with other molecules or environments [85].

The placement and type of the negatively charged residues throughout the protein sequence determine how two proteins with 55 and 21 negatively charged residues differ from one another. To be more precise, these charged residues can be negatively charged (like glutamic acid, aspartic acid) or positively charged (like lysine, arginine). It is possible that the protein with 55 charged residues has a greater net charge than the protein with 21 charged residues. Assume that most of these residues have a positive charge. If the protein is primarily negatively charged, the net charge will be negative; otherwise, the protein will have an overall positive net charge. The balance between positively and negatively charged residues affects the net charge of a protein at a specific pH. A higher positive net charge would arise from a greater quantity of positively charged residues. A greater negative net charge would arise from a greater quantity of negatively charged residues.

As discussed, the pI of a protein is the pH at which it carries no net electrical charge. The distribution of charged residues affects the pI value. Proteins with more positively charged residues typically have a higher pI, whereas proteins with more negatively charged residues tend to have a lower pI. Therefore, the protein with 55 charged residues might have a different pI compared to the protein with 21 charged residues, depending on the distribution of these residues and their specific pKa values. Proteins with varying numbers of charged residues may interact differently with other molecules or surfaces. For instance, a protein with many positively charged residues might interact more strongly with negatively charged molecules or surfaces.

In contrast, a protein with many negatively charged residues might interact more strongly with positively charged molecules or surfaces. The distribution and number of charged residues can also influence the protein’s biological function. For example, proteins with many positively charged residues might be involved in DNA binding. In contrast, proteins with many negatively charged residues might participate in interactions with RNA or other negatively charged molecules.

The total of two particular amino acids, arginine (Arg) and lysine (Lys), is referred to as the “Total number of positively charged residues (Arg + Lys)”. Because these amino acids have amino groups that may take up a proton (H+) in solution and form a positively charged amino group (NH3+), these amino acids are positively charged. Depending on the pH of the surrounding environment, arginine and lysine contribute to the overall positive charge of a protein molecule. These residues typically have a positive charge and are protonated at pH values lower than their corresponding pKa values, which indicate the pH at which 50% of the molecules are protonated. They typically contain no net charge and are deprotonated at pH levels higher than their pKa values. Understanding the overall charge distribution of a protein requires knowledge of its total amount of positively charged residues (Arg + Lys). It can affect the behavior of the protein at different pH levels, as well as a range of biological processes and interactions with other molecules. Proteins with a high concentration of positively charged residues may interact more favorably with surfaces or molecules that are negatively charged. Proteins with a high concentration of negatively charged residues, on the other hand, can interact more favorably with positively charged surfaces or molecules. In conclusion, a protein’s charge distribution and possible interactions with other molecules or surroundings can be inferred from the total number of positively charged residues (Arg + Lys).

The main differences between the two proteins with 55 and 23 positive charged residues (Arg + Lys) are the overall positive charge distribution and possible interactions. This is where the difference could show up: compared to a protein with 23 positively charged residues, the protein with 55 positively charged residues will probably have a higher net positive charge. The behavior and interactions of the protein may be significantly affected by this increased net positive charge, particularly in situations where negatively charged molecules or surfaces are present. A protein’s distribution and quantity of positively charged residues impact its isoelectric point or pI. A higher pI is typically found in proteins with a greater number of positively charged residues.

Consequently, compared to a protein with 23 positively charged residues, the protein with 55 positively charged residues may have a larger pI. Positively charged residues in proteins may enhance their interaction with negatively charged molecules or surfaces. These contacts might involve attaching to negatively charged membranes, interacting with negatively charged areas of other proteins, or binding to nucleic acids (DNA or RNA). Because of its higher net positive charge, the protein with 55 positively charged residues may interact with negatively charged molecules or surfaces more strongly than the protein with 23 positively charged residues. The quantity and distribution of positively charged residues can affect how a protein functions biologically. Proteins with many positively charged residues may be involved in membrane association, enzymatic activity, or DNA or RNA binding. The overall structure, additional amino acid residues, and the cellular environment in which the proteins with 55 and 23 positively charged residues function will determine their particular roles. In conclusion, differences in the positive charge distribution of two proteins can affect their interactions, stability, and biological functions. These variations are indicated by the difference in the total amount of positively charged residues (Arg + Lys) between the two proteins.

Furthermore, a protein’s total number of negatively charged residues plays a crucial role in its behavior and function. Negatively charged residues, such as aspartic acid (Asp) and glutamic acid (Glu), contribute to the overall net charge of a protein. These charges help prevent protein aggregation by maintaining solubility. Protein aggregation can lead to dysfunction or disease, while solubility is essential for proper protein folding, interactions, and cellular processes. Charged residues form ion pairs, hydrogen bonds, and other electrostatic interactions. These interactions influence protein structure, folding, binding, and condensation. Long-range electrostatic effects impact protein behavior, including ligand binding and enzymatic reactions. As proteins are synthesized, the nascent polypeptide passes through the negatively charged exit tunnel of the ribosome; positively charged stretches within the nascent peptide can interact with ribosome walls and slow down translation. Thus, charged polypeptides affect protein expression and translation efficiency. Charge ladders involve chemical modification of charged residues to generate derivatives with varying charges [86].

The estimated half-life of a protein refers to the time it takes for half of the protein molecules in a cell or biological system to be degraded or otherwise become inactive. Protein half-life can vary widely depending on several factors, including the specific protein, cell type, organism, and physiological conditions. In general, the half-life of proteins can range from minutes to days or even longer. Some proteins have very short half-lives, meaning they are rapidly turned over within cells, while others are more stable and persist for more extended periods. For example, (a) short-lived proteins, which are involved in cellular signaling, regulation, or response to environmental changes, often have short half-lives. These proteins are rapidly synthesized and degraded as part of the cell’s dynamic response to stimuli. Examples include transcription factors, cell cycle regulators, and specific signaling molecules. (b) Long-lived proteins are structural proteins, enzymes, and proteins that maintain cellular structure and function and tend to have longer half-lives. These proteins are essential for the cell’s structure and function and are typically turned over more slowly. Examples include structural components of the cytoskeleton, enzymes involved in primary metabolic processes, and histones [87,88]. 

The half-life of a protein is influenced by various factors: (a) Protein structure: proteins with specific structural features, such as disordered regions or post-translational modifications, may be more susceptible to degradation. (b) Cellular environment: cellular conditions such as nutrient availability, stress, and signaling pathways can affect protein stability and turnover rates. (c) Protein interactions: protein–protein interactions and association with other cellular components can influence protein stability and degradation. (d) Post-translational modifications: modifications such as ubiquitination or phosphorylation can target proteins for degradation by the proteasome or lysosomes, affecting their half-life. Estimating the half-life of a specific protein often involves experimental approaches such as pulse–chase assays, metabolic labeling, or computational modeling. These techniques help researchers understand protein turnover dynamics and their roles in cellular processes. Additionally, databases and computational tools provide estimates or predictions of protein half-lives based on experimental data and computational algorithms, aiding researchers in studying protein dynamics and cellular regulation. Overall, inhibitory proteins have a half-time of five times greater than stimulant ones. Their biological effect lasts longer and is less susceptible to degradation than stimulant proteins. 

The instability index of a protein is a numerical value that predicts the stability of a protein based on its amino acid sequence. It was introduced by Guruprasad et al. in 1990 as a method to estimate the stability of proteins from their primary sequence. The instability index is calculated using a formula that considers various physicochemical properties of amino acids in the protein sequence, including the relative volume of each amino acid, the hydropathy index, and the presence of dipeptides that tend to occur in unstable regions. The instability index can be helpful for researchers in various areas, including protein engineering, protein expression, and structural biology. It provides a quick and rough estimate of a protein’s stability based solely on its amino acid sequence, which can help researchers prioritize proteins for further study or experimental manipulation. However, it is important to note that the instability index is just one of many factors that contribute to protein stability, and experimental validation is often necessary to confirm the predicted stability of a protein. The instability index is computed after the following formula: 

Instability index = 10 × (*N*total large + *n*charged − length total), where n large is the number of amino acids with high relative volume (Val, Ile, Leu, Phe, Tyr, and Trp), n charged is the number of charged amino acids (Arg, Lys, Asp, and Glu), N total is the total number of amino acids in the sequence, and length is the length of the protein sequence. Previous studies have shown that both proteins are stable [89,90,91].

The aliphatic index of a protein is a measure of its thermostability, specifically related to the aliphatic amino acids present in its sequence. Aliphatic amino acids are those with non-aromatic side chains, which typically include alanine (Ala), valine (Val), isoleucine (Ile), and leucine (Leu). The aliphatic index is calculated based on the relative volume occupied by aliphatic side chains in the protein, contributing to its stability at high temperatures. A higher aliphatic index suggests a more significant proportion of aliphatic amino acids in the protein sequence, which is associated with increased thermostability. The difference in aliphatic index between the two proteins is the following: the inhibitory chimeric model has an index of 86.32. This protein has a high aliphatic index, indicating a significant proportion of aliphatic amino acids in its sequence. Such proteins are typically more stable at high temperatures and may be better adapted to environments with extreme conditions, such as heat or pH extremes. The stimulant chimeric model has an aliphatic index of 54.88—this suggests a lesser proportion of aliphatic amino acids in its sequence, which may result in lower thermostability than the protein with the higher aliphatic index.

In summary, the difference in aliphatic index between these two proteins suggests differences in their potential thermostability. The protein with the higher aliphatic index (86.32) is likely more thermostable than the protein with the lower aliphatic index (54.88). However, other factors beyond aliphatic amino acids, such as overall protein structure and composition, can also influence a protein’s stability [92,93].

The grand average of hydropathicity (GRAVY) is a measure that quantifies the overall hydrophobicity or hydrophilicity of a protein sequence. It is calculated by averaging the hydropathy values of all amino acids in the sequence. Hydropathy values represent the relative hydrophobicity or hydrophilicity of amino acids. Positive hydropathy values indicate hydrophobic amino acids (which tend to be buried inside the protein structure away from water). In contrast, negative values indicate hydrophilic amino acids (those that tend to be exposed to the aqueous environment). The GRAVY score is calculated by summing the hydropathy values of all amino acids in the sequence and dividing by the number of residues. A negative GRAVY score indicates a predominance of hydrophilic residues in the protein sequence, while a positive GRAVY score indicates a predominance of hydrophobic residues. The inhibitory chimeric model has a GRAVY score of −0.258; this protein has a negative GRAVY score, suggesting that, on average, its amino acid sequence is hydrophilic. Such proteins will likely have more polar or charged residues on their surface, making them more soluble and potentially interacting favorably with water molecules.

With a GRAVY score of −0.594, the stimulant chimeric model’s protein is even more hydrophilic than the first protein, indicating a lower GRAVY score. Its sequence probably has more hydrophilic residues than the protein, with a GRAVY value of −0.258. In conclusion, variations in the GRAVY scores of these two proteins point to variations in their general hydrophilicity. Compared to the protein with the higher GRAVY score (−0.258), the one with the lower value (−0.594) is probably even more hydrophilic [94]. 

Comprehending the molecular architecture of a protein is crucial for deciphering the correlations between its structure and function, forecasting its biological functions, and developing ligands or modulators that engage with particular protein sections or characteristics. Computational approaches, structural biology methods (such as X-ray crystallography and nuclear magnetic resonance spectroscopy), and bioinformatics tools for sequence and structural analysis can all be used to analyze the chemical space of proteins. Both spaces have the same geometry by comparing the inhibitory and stimulant proteins and chemical space. The inhibitory space is narrower than the stimulant one. Also, the stimulant space is more represented in the negative domain, whereas the inhibitory space occupies both negative and positive domains. These results are based on the chemical space representation by chemical descriptors, which follows the chemical space represented by polynomial equations.

The chimeric models’ chemical spaces both show a dimensional reduction, as expected. Both spaces have the same geometry. In opposition to the protein chemical spaces, the chimeric model space is more expansive than the stimulant chimeric model. 

The “C-alpha distance map” explicitly shows the distances between C-alpha atoms and often depicts the spatial arrangement of atoms in a protein structure. The C-alpha atom, a component of the protein’s backbone, is utilized in protein structure as a point of reference to characterize the general folding pattern. The distances between each pair of C-alpha atoms in a protein structure are shown graphically in the C-alpha distance map. This map can be used to comprehend the spatial interactions between various protein components, detect structural motifs, and examine the overall folding pattern [95]. 

All three polynomials are of degree 6. The leading coefficients are inhibitory space: −23.758, stimulant space: −1.1017, and combined space: −7.9346. The behavior is determined by the leading term of the polynomial: inhibitory space: As *x*→±∞x→±∞, *y*→−∞y→−∞, stimulant space; as *x*→±∞x→±∞, *y*→−∞y→−∞, combined space; as *x*→±∞x→±∞, *y*→−∞y→−∞.While all three polynomials have the same degree, their leading coefficients and coefficients of the other terms differ, leading to distinct behaviors and shapes. 

The leading coefficient in the equation generated from inhibitory space is negative (−23.758), meaning that the polynomial function both increases and reduces quickly as x increases and lowers. The function’s general shape is likewise influenced by the other coefficients. For example, the positive coefficient of *x*4x4 implies that there can be local maxima and minima for the function. Because the coefficients’ signs alternate, the function may behave oscillatorily or have several turning points. The function approaches negative infinity as x approaches either positive or negative infinity, showing a decreasing tendency at both extremes. Compared to the other two functions, the leading coefficient (−23.758) indicates a stronger decreasing trend.

In the stimulant space, similar to the inhibitory space, the leading coefficient is negative (−1.1017), indicating a downward trend at both extremes. The coefficients contribute to the shape of the function. For example, the positive coefficient of *x*4 suggests the presence of local maxima and minima. The function may also exhibit oscillatory behavior or have multiple turning points. As x approaches positive or negative infinity, the function approaches negative infinity. The leading coefficient is less negative (−1.1017), indicating a relatively less steep downward trend than the inhibitory space.

Finally, a downward trend is indicated at both extremities by the negative (−7.9346) leading coefficient in the combined space function. Local maxima and minima may result from the coefficients’ influence on the function’s form. Similar to other functions, there could be several turning points or oscillatory behavior. The function becomes closer to negative infinity as x gets closer to positive or negative infinity. Although it is likewise negative (−7.9346), the leading coefficient’s size places it in between the other two, indicating an intermediate rate of decline.

All three polynomial functions exhibit a downward trend at both extremes, with potential oscillatory behavior and multiple turning points. The specific values of the coefficients will determine each function’s exact shape and behavior. Graphing these functions would provide a more precise visualization of their behavior and any distinctive features they may have.

Each polynomial has different coefficients for terms of higher orders (i.e., x4, x5, x6). These coefficients contribute to the shape of the polynomial curve and influence the presence of local extrema (maxima and minima). The inhibitory space has more significant magnitude coefficients for most higher-order terms than the other two, potentially leading to more pronounced oscillations or sharper turns in the curve. Stimulant space and combined space have more moderate coefficients for higher-order terms, suggesting smoother curves than inhibitory space.

Critical points, where the function’s derivative is zero, correspond to potential local extrema or inflection points. The locations and nature of these vital points would depend on the specific values of the coefficients in each polynomial. Due to its unique coefficient values, inhibitory space might have critical points at different locations than stimulant and combined space. Inhibitory space may exhibit more erratic behavior than smoother curves of stimulant space and combined space, given its more significant and steeper leading coefficients (as seen in the figure above).

Overall, regarding the inhibitory space, this polynomial function might represent a scenario where the response or activity is inhibited or suppressed. Inhibitory processes are standard in various biological and physical systems where certain factors decrease the activity or effectiveness of other factors. Multiple roots, critical points, and inflection points suggest a complex behavior with potential oscillations or fluctuations in the inhibitory response. The negative leading coefficient indicates a downward trend, suggesting that as the input *x* increases, the inhibitory effect becomes more robust, decreasing the response or activity.

Stimulant space—this polynomial function may represent a scenario where the response or activity is stimulated or enhanced. Stimulant processes are often observed in biological, chemical, and physical systems where certain factors increase the activity or effectiveness of other factors. Like the inhibitory space, multiple roots, critical points, and inflection points suggest a complex behavior with potential oscillations or fluctuations in the stimulant response. The negative leading coefficient also indicates a downward trend, suggesting that the stimulant effect strengthens as the input x increases, increasing the response or activity.

The combined space polynomial function combines elements of both inhibitory and stimulant effects, perhaps representing a scenario where both factors simultaneously influence the overall response or activity. Multiple roots, critical points, and inflection points suggest a complex interaction between inhibitory and stimulant processes, leading to potentially intricate behavior. The negative leading coefficient indicates an overall downward trend, but the specific behavior depends on the combined effects of the individual terms in the polynomial.

Overall, these polynomial functions provide mathematical representations of complex processes in inhibitory, stimulant, and combined spaces. Their analysis helps understand the behavior and interactions of factors within these spaces. It can be valuable in various fields, such as biology, chemistry, physics, and economics.

In the context of angiogenesis, the inhibitory space polynomial function might represent factors or processes that inhibit or suppress angiogenesis. The polynomial’s complex behavior, with multiple roots, critical points, and inflection points, could represent the intricate interplay of various inhibitory factors in regulating angiogenesis. For example, specific molecules like angiostatin or Endostatin inhibit angiogenesis by blocking the activity of pro-angiogenic factors. The polynomial could represent the combined effect of these inhibitory factors.

In the context of angiogenesis, the stimulant space polynomial function might represent factors or processes that stimulate or promote angiogenesis. Like the inhibitory space, the polynomial’s complex behavior could represent the multifaceted nature of stimulatory factors in regulating angiogenesis. For instance, Vascular Endothelial Growth Factor (VEGF) and Fibroblast Growth Factor (FGF) are potent angiogenesis stimulators. The polynomial could represent the combined effect of these stimulatory factors.

The combined space polynomial function combines inhibitory and stimulant effects on angiogenesis. In the context of angiogenesis, this polynomial could represent the balance between inhibitory and stimulatory factors that determine the net impact on angiogenesis. The polynomial’s behavior reflects the complex interactions between factors that promote or inhibit angiogenesis, resulting in intricate regulation of blood vessel formation.

## 4. Materials and Methods

To explore the chemical space of VEGFR2, ligand docking and protein–protein docking methodologies were used. Firstly, VEGFR2 was energetically minimized and protonated at physiological pH and temperature. AMBBER 99 force field was used for all protein preparation and docking computations. Regarding the ligand docking, a set of 278 molecules were retrieved randomly from the ChEMBL 01 database using a randomized extraction protocol using the MtiOpneScreen server [96,97]. The molecules were energetically minimized and protonated at physiological pH and temperature. The VEGFR2-binding site was computed using MOE 2009 software and from the literature [98]. AutoDock software was used to dock the ligands [99]. Docking results were selected based on the total energy of the complex (Kcal/mol). The total energy, solvation energy, and angular energy were computed for each protein–ligand complex. The first 27 energetically favorable ligands are represented in Appendix A (the rest of the 278 ligands are described in Appendix A).

Regarding protein–protein docking, a series of compounds with demonstrated antiangiogenetic or angiogenetic activity were selected from the literature to explore the chemical space. The preferred compounds were studied computationally. The following molecules were chosen as angiogenesis stimulators: Insulin-Like-Growth-Factor-1 (IGF-1) (PDB ID 1B9G) [100], Basic Fibroblast Growth Factor (bFGF) (PDB ID 1BFB) [101], Hepatocyte Growth Factor (HGF) (PDB ID 1GP9) [102], Human Epidermal Growth Factor (EGF) (PDB ID 1JL9) [103], Transforming Growth Factor Beta 1 (TGF beta-1) (PDB ID 1KLA) [104], Human Platelet-Derived Growth Factor Bb (PDGF B) (PDB ID 1PDG) [105], Angiopoietin 2 (PDB ID 1Z3S) [106], Human Vascular Endothelial Growth Factor-B (VEGFB) (PDB ID 2C7W) [107], Human Transforming Growth Factor Alpha TGF alpha (PDB ID 2TGF) [108], Vascular Endothelial Growth Factor C (VEGFC) (PDB ID 2X1W) [109], Vascular Endothelial Growth Factor D (VEGF D) (PDB ID 2XV7) [110], Interleukin 8 (IL8) (PDB ID 3IL8) [111], Platelet-Derived Growth Factor A (PDGF A) (PDB ID 3MJK) [112], Angiopoietin 1 (PDB ID 4JYO) [113], Human Transforming Growth Factor Alpha (TNF alpha) (PDB ID 4TGF) [114], Vascular Endothelial Growth Factor A (VEGFA) (PDB ID 6Z13) [115], Platelet-Derived Growth Factor C (PDGFC) (homology model 1 UniProt ID Q9NRA1) [116], Phosphatidylinositol-Glycan Biosynthesis Class F Protein (PIGF) (homology model 2 UniProt ID Q07326) [117], and Platelet-Derived Growth Factor D (PDGFD) (homology model 3, UniProt ID Q9GZP0) [118]. Molecules that were chosen as inhibitors are the following: Human Interleukin-4 (IL4) (PDB ID 1BBN) [119], Human Interleukin-12 (IL12) (PDB ID 1F45) [118], Interferon-gamma (PDB ID 1HIG) [120], Human Pigment Epithelium-Derived Factor (PEDF) (PDB ID 1IMV) [121], Human Angiostatin (PDB ID 1KI0) [122], Endostatin (PDB ID 1KOE) [123], Thrombostatin 1 (PDB ID 1LSL) [124], Human Interferon Alpha (PDB ID 1RH2) [125], Human Skeletal Muscle Troponin (PDB ID 1YTZ) [126], Thrombospondin 2 (PDB ID 2RHP) [127], antithrombin II (PDB ID 4EB1) [128], and Vasostatin (homology model 4 UniProt ID P10645) [129]. Protein–protein docking was performed using the HADDOCK 2.0 server [130]). The total, solvation, and angular energy were computed for each protein–protein complex. To explore the chemical space of inhibitors and stimulants of tyrosine kinase concerning angiogenesis, molecular descriptors were calculated using ChemDes(Web) software packages [131,132,133] for all proteins (inhibitory and stimulants). Using molecular descriptors (number of H bond acceptors, number of H bond donors, polar surface area, shape attribute, the sum of degrees, sum of valence degrees), the chemical space for VEGFR stimulators and inhibitors was characterized and represented as radial graphs. Furthermore, the chemical space was computed using the same methodology for the chimeric models. Also, docking of the chimeric models with VEGFR 2 was performed, and the total energy, solvation energy, and angular energy were calculated (kcal/mol). To further explore the molecular systems of angiogenesis stimulants and inhibitors, c-c atom distances were computed. Based on the carbon–carbon distance matrix, a multidimensional space was represented. Based on the multifaceted space representation of considerable data reduction, a six-degree polynomial equation system was computed for the chimeric and inhibitory chimeric models. The six-degree equations calculated 2D and 3D space maps using the online computational server Wolphram Alpha [134]. Ramachandran plots were also used to assess the stability and reliability of the protein models. Finally, Aa sequences and molecular descriptor data were compared to obtain insights into angiogenesis’s chemical stimulant and inhibitor space. 

This study provides an overview of the main features of molecules that either inhibit or stimulate angiogenesis. The findings of the study can be utilized to create a potent stimulator of angiogenesis. Both wet lab experiments and computational methods are necessary to accomplish this objective. The resulting molecular systems can then be employed to develop a pharmacologically active stimulator of angiogenesis, which can target either inhibitors or multiple targets simultaneously.

## 5. Conclusions

The chemical space of angiogenesis stimulators and inhibitors is slightly similar. However, the chemical space of inhibitors is more expended than stimulators, indicating a most probable interaction. A most probable interaction with the inhibitor space is due to the inhibitors’ expenses of the chemical space being more conformationally favorable for a diverse set of molecules compared to the stimulants. Also, a broader chemical space is more energetically and conformationally favorable than a less expanded chemical space. These characteristics are also transposed to the chimeric models, where the inhibitor chimeric model is a larger-size molecule than the stimulant chimeric model. Also, regarding the molecular interactions, the inhibitors have slightly more favorable complex energies than the stimulants. Mathematically, the inhibitory space has a narrower domain than the stimulant space but expands in negative and positive domains. This means the interactions are possible with distinct and variated conformations compared to the stimulant space. Also, interaction with molecules that pose symmetry is favorable. The stimulant space is expended mainly on the negative larger domain. The consequence of this geometry is primarily a selective, wider domain for more specific and less accessible conformations. 

The chemical space and domain distribution are critical factors in VEGFR2 behavior as a stimulant or angiogenesis inhibitor. Further experimental and in silico studies are needed to characterize and quantify the complex VEGFR system and its role in angiogenesis and vascular morphogenesis.

## Figures and Tables

**Figure 1 ijms-25-07787-f001:**
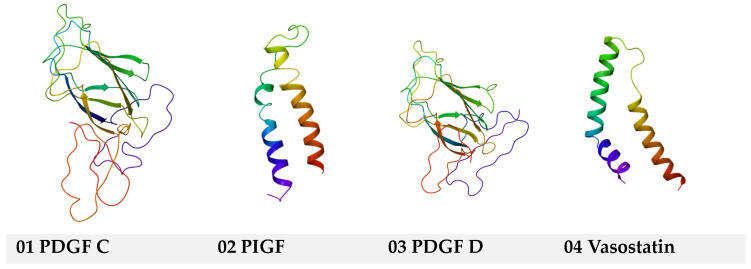
Homology models of PDGF C, PIGF, PDGF D, and Vasostatin.

**Figure 2 ijms-25-07787-f002:**
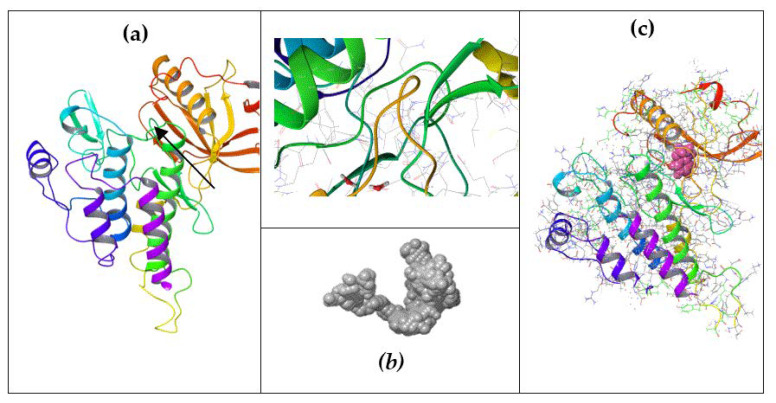
(**a**) VEGFR2 is represented as a ribbon; cavity 1 is defined as water clusters (molecules of water shown in grey); the coordinates of the binding site are shown by a black arrow (x = 29.04 Å; y = −36.68 Å; z = −18.54 Å); details of the binding site are also represented; (**b**) binding site of VEGFR2 detail and binding site space filling; (c) VEGFR2 docked with 2H-chromen-2-one (slightly moved compared to (**a**) to show the ligand—colored in pink—in the binding pocket).

**Figure 3 ijms-25-07787-f003:**
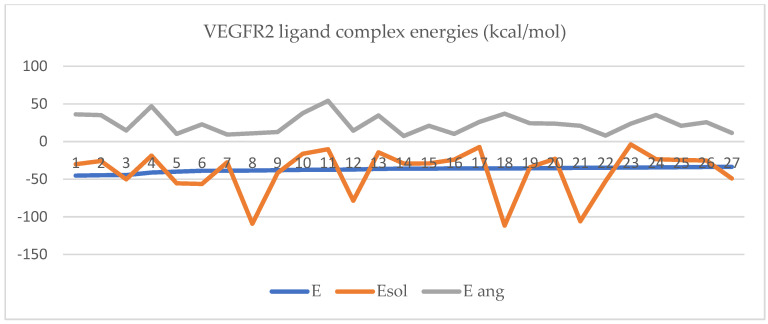
Docking energies of VEGFR2 against a set of Chembl1 structures. For ligands 1–27, see Appendix A and Section 4. E_total energy (kcal/mol); E sol—solvation energy (Kcal/mol); E ang—angulation energy (kcal/mol) (ligand names and structures can be found in Appendix A).

**Figure 4 ijms-25-07787-f004:**
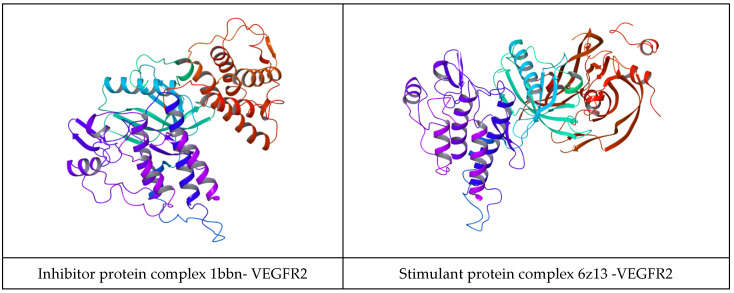
The protein–protein complex of an inhibitory and stimulant protein is docked with VEGFR2 represented in ribbons.

**Figure 5 ijms-25-07787-f005:**
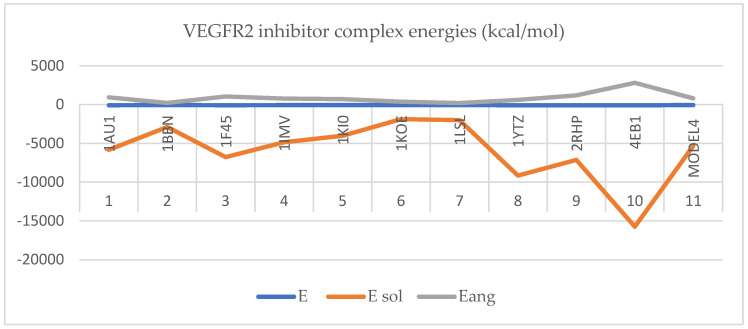
VEGFR2 inhibitor complex energies (kcal/mol): E−the total complex energy; E sol—solvation energy; E ang—angular energy.

**Figure 6 ijms-25-07787-f006:**
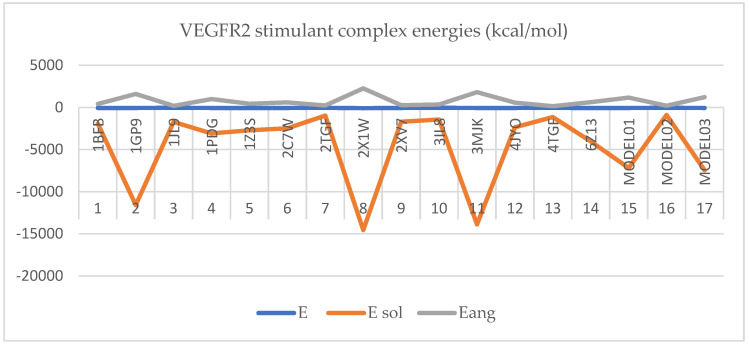
VEGFR2 stimulator complex energies (kcal/mol). E ref—the total complex energy; E sol—−solvation energy; E ang—angular energy.

**Figure 7 ijms-25-07787-f007:**
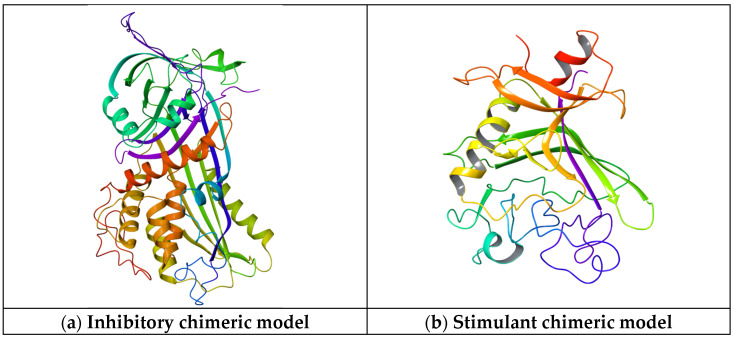
Chimeric protein models for the inhibitory and stimulant proteins.The models are represented as ribbons.

**Figure 8 ijms-25-07787-f008:**
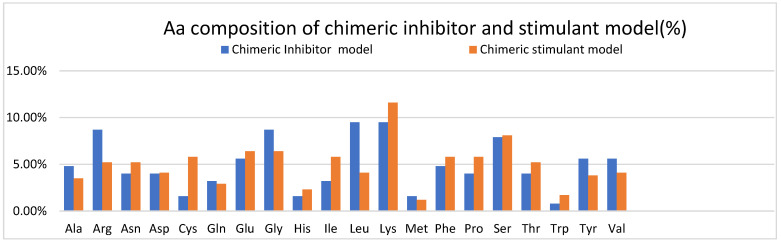
Aa composition (%) of inhibitory and stimulant chimeric models.

**Figure 9 ijms-25-07787-f009:**
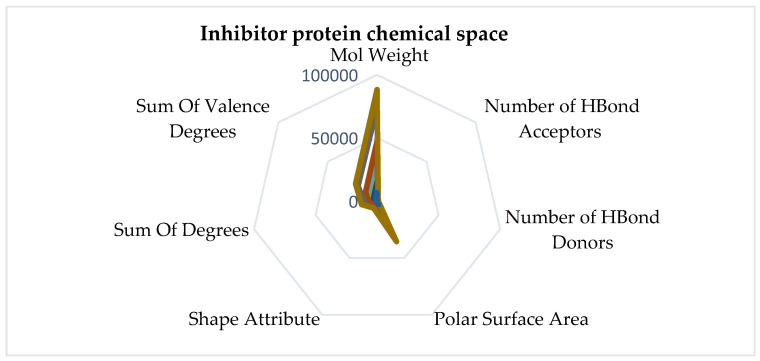
Chemical space of inhibitory and stimulant proteins of angiogenesis. The chemical space is characterized by the six molecular descriptors: mol weight, number of H bond acceptors, number of H bond donors, polar surface area, shape attribute, sum of degrees, and sum of valence degrees. The chemical space is represented as radar plots.

**Figure 10 ijms-25-07787-f010:**
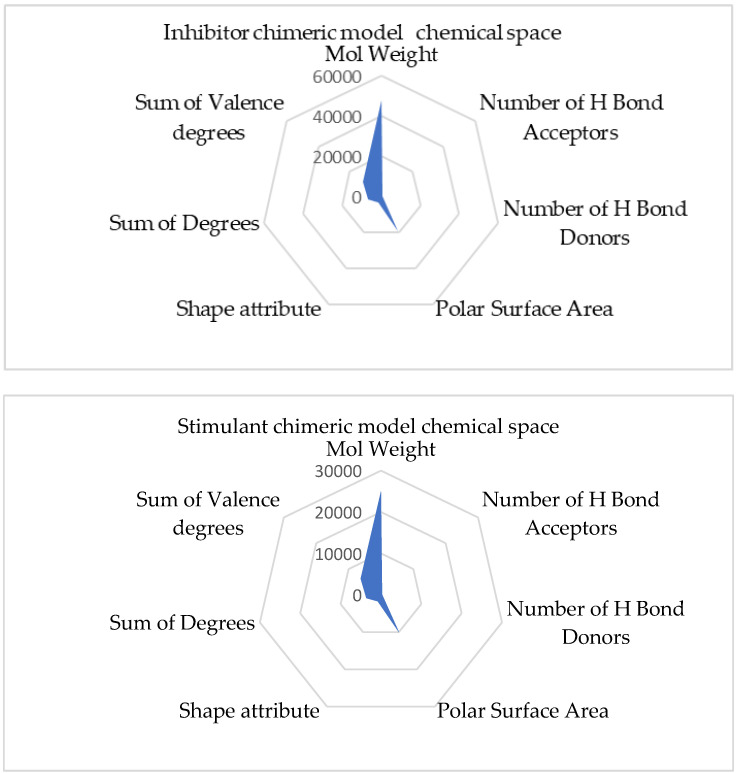
Angiogenesis chemical space of inhibitory and stimulating proteins in chimeric models characterized by molecular weight, number of hydrogen bond donors, number of hydrogen bond acceptors, shape attribute of each molecule, polar surface, and the sum of degrees.

**Figure 11 ijms-25-07787-f011:**
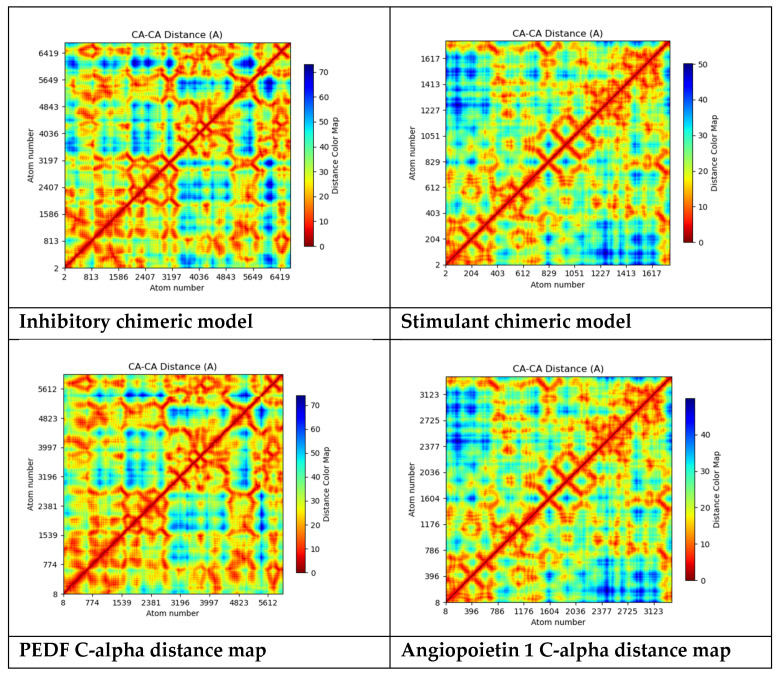
C-alpha distance map plot of inhibitory and stimulant chimeric models together with PEDF (as an inhibitor example) and Angiopoietin 1 (as a stimulant example).

**Figure 12 ijms-25-07787-f012:**
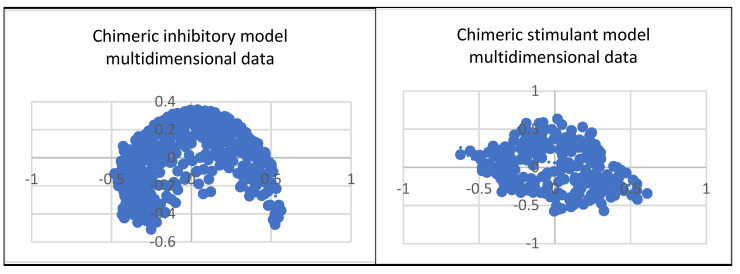
Chimeric and stimulant model multidimensional data represented as scatter plots.

**Figure 13 ijms-25-07787-f013:**
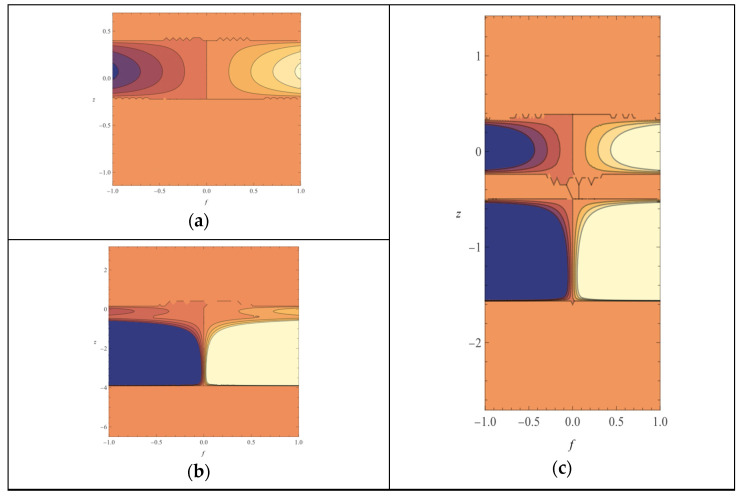
Two-dimensional complex map of the six-degree polynomial equation: (**a**) inhibitory space equation; (**b**) stimulant space equation; (**c**) combined space equation.

**Figure 14 ijms-25-07787-f014:**
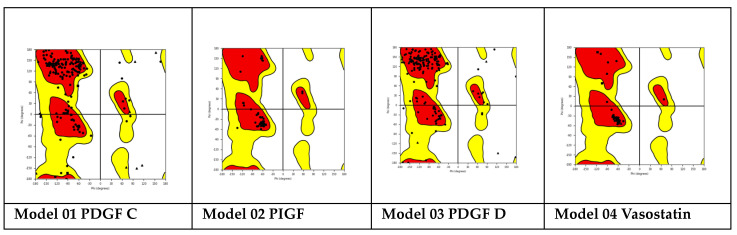
Ramachandran plots for the homology models used in the computations.

**Table 1 ijms-25-07787-t001:** Chimeric proteins’ Aa sequences.

INHIBITORY CHIMERIC MODEL	STIMULANT CHIMERIC MODEL
10 20 30 40 50 60 VDICTAKPRD IPMNPMCIYR SPENRRVWEL SKANSRFATT FYQHLADSKN DNDNIFLSPL 70 80 90 100 110 120 SISTAFAMTK LGACNDTLQQ LMEVFKFDTI SEKTSDQIHF FFAKLNCRLY RKANKASKLV 130 140 150 160 170 180 SANRLFGDKS LTFNETYQDI SELVYGAKLQ PLDFKENAEQ SRAAINKWVS NKTEGRITDV 190 200 210 220 230 240 IPSEAINVLV LVNTRTSTVL VLVNTIYFKG LWKSKFSPEN TRKELFYKAD GESCSASMMY 250 260 270 280 290 300 QEGKFRYRRV AEGTQVLELP FKGDDITMVL ILPKPEKSLA KVEKELTPEV LQEWLDELEE 310 320 330 340 350 360 MMLVVHMPRF RIEDGFSLKE QLQDMGLVDL FSPEKSKLPG IVAEGRDDLY VSDAFHKAFL 370 380 390 400 410 EVNEEGSEAA ASTAVVIAGR SLNPNRVTFK ANRPFLVFIR EVPLNTIIFM GRVANPCVK	10 20 30 40 50 60 PFRDCADVYQ AGFNKSGIYT IYINNMPEPK KVFCNMDVNG GGWTVIQHRE DGSLDFQRGW 70 80 90 100 110 120 KEYKMGFGNP SGEYWLGNEF IFAITSQRQY MLRIELMDWE GNRAYSQYDR FHIGNEKQNY 130 140 150 160 170 180 RLYLKGHTGT AGKQSSLILH GADFSTKDAD NDNCMCKCAL MLTGGWWFDA CGPSNLNGMF 190 200 210 YTAGQNHGKL NGIKWHYFKG PSYSLRSTTM MIRPLDF

**Table 2 ijms-25-07787-t002:** Chimeric model properties.

Property	Inhibitory Chimeric Model	Stimulant Chimeric Model
Number of amino acids	419	217
Molecular weight	47,642.80	24,944.16
Theoretical pI	7.06	8.32
Total number of negatively charged residues (Asp + Glu)	55	21
Total number of positively charged residues (Arg + Lys)	55	23
Formula	C2135H3376N568O629S18	C1115H1648N304O320S16
Total number of atoms	6726	3403
Estimated half-life	100 h (mammalian reticulocytes, in vitro). >20 h (yeast, in vivo). >10 h (Escherichia coli, in vivo).	>20 h (mammalian reticulocytes, in vitro). >20 h (yeast, in vivo). ? (Escherichia coli, in vivo).
Instability index:	The instability index (II) is computed to be 38.35 This classifies the protein as stable.	The instability index (II) is computed to be 35.30 This classifies the protein as stable.
Aliphatic index	86.32	54.88
Grand average of hydropathicity (GRAVY)	−0.258	−0.594

## Data Availability

Data is contained within the Appendix A.

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
