# Peer review of "Variations of VEGFR2 Chemical Space: Stimulator and Inhibitory Peptides"

_ijms, 2024, doi:10.3390/ijms25147787_

Round 1
Reviewer 1 Report
Comments and Suggestions for Authors
The manuscript entitled “Variations of VEGFR2 Chemical Space: Stimulator and Inhibitory Peptides” describes an in silico examination of the molecular properties of VEGFR2, a molecule primarily involved in the processes of vasculogenesis and angiogenesis. It is a well-written manuscript with a lot of modeling studies but there are some points need to be considered:
1-Extensive research has categorized various pro-angiogenic molecules, including angiogenic proteins, gene therapy, peptide drugs, and organic molecules. Give an example for each class with a brief description of its chemical structure.
2-The whole manuscript should be reviewed for correction editing and typing mistakes like:
Line 118: Correct --12.71 to -12.71
Put the reference No. 13 before the dot not after.
Line 493: The references [35,36,37] corrected to [35-73]
The title of table 1 should be placed above the table not below it.
3- Write a short paragraph before the conclusion discussing the future prospects of this study
4- References are written with a wrong format. The year should be in bold without bracket and the name of the journal should be italic.
5-The author said there is a supplementary file S1 while there was no attached supplementary file and the link for supplementary materials didn’t work.
Comments on the Quality of English LanguageThe English language is fine except some few mistakes need correction.
Author Response
Reviewer 1
The manuscript entitled "Variations of VEGFR2 Chemical Space: Stimulator and Inhibitory Peptides" describes an in silico examination of the molecular properties of VEGFR2, a molecule primarily involved in the processes of vasculogenesis and angiogenesis. It is a well-written manuscript with a lot of modeling studies but there are some points need to be considered:
Comment1: 1-Extensive research has categorized various pro-angiogenic molecules, including angiogenic proteins, gene therapy, peptide drugs, and organic molecules. Give an example for each class with a brief description of its chemical structure.
Response1: The injtroduction part was correc ted as suggested. The foloowewing text was added togheter with references.: Angiogenic proteins are essential molecules involved in angiogenesis, forming new blood vessels from existing ones. This mechanism is crucial for various physiological and pathological situations, including wound healing, embryonic development, and tumor growth. Examples of such proteins are Vascular Endothelial Growth Factor (VEGF), Basic Fibroblast Growth Factor (bFGF), Angiopoietins (Ang-1 and Ang-2), Platelet-Derived Growth Factor (PDGF ), Transforming Growth Factor-Beta (TGF-β), Epidermal Growth Factor (EGF), Hepatocyte Growth Factor (HGF), Matrix Metalloproteinases (MMPs). Re-garding their structure, VEGF is a dimeric glycoprotein, often consisting of two identical monomers linked by disulfide bonds. The primary isoforms (e.g., VEGF-A) have variations in their sequence due to alternative splicing.
bFGF is a single-chain polypeptide with a compact, globular structure. It has a high affinity for heparan sulfate proteoglycans, which stabilize it and enhance its signaling. Angiopoietins are secreted glycoproteins with a characteristic structure, including an N-terminal superclustering domain, a central coiled-coil domain for dimerization, and a C-terminal fibrinogen-like domain for receptor binding. PDGF is a dimeric protein consist-ing of A and B chains that can form homo- or heterodimers (e.g., PDGF-AA, PDGF-BB, PDGF-AB). Disulfide bonds link the monomers. TGF-β is a dimeric peptide, with each monomer having a cystine knot motif, a common feature among the TGF-β superfamily. The dimer is stabilized by disulfide bonds between the monomers. EGF is a small poly-peptide consisting of 53 amino acids with three intramolecular disulfide bonds that create a compact, stable structure. The presence of these disulfide bonds is critical for its biologi-cal activity. HGF is a large, heterodimeric protein composed of an alpha chain (69 kDa) and a beta chain (34 kDa) linked by a single disulfide bond. The alpha chain contains four kringle domains and an N-terminal hairpin domain, while the beta chain has serine pro-tease homology. MMPs are a family of zinc-dependent endopeptidases. They have a common structure consisting of a pro-domain (which maintains them in an inactive form), a catalytic domain with a zinc-binding motif, a hinge region, and a hemopexin-like C-terminal domain that contributes to substrate specificity and interaction with tissue in-hibitors[18-26].
Regarding gene therapy, VEGF Geme Therapy, FGF Gene Therapy, HGF Gene Thera-py, Angiopoietin Gene Therapy, PDGF Gene Therapy, and combined gene therapy, respec-tively, are optimal angiogenesis solutions at various stages of research and clinical devel-opment, with some having shown promising results in early-phase clinical trials[27,28].
The most studied form of VEGF gene therapy, involving the delivery of the VEGF-A gene to promote the formation of new blood vessels. Clinical trials have tested its efficacy in treating peripheral arterial disease and myocardial ischemia. VEGF-C and VEGF-D are targeted for lymphangiogenesis and angiogenesis. These genes have been explored for treating lymphedema and enhancing wound healing. FGF-1 (Acidic Fibroblast Growth Factor) delivers the FGF-1 gene, aims to enhance angiogenesis, and has been tested in clin-ical trials for treating coronary and peripheral artery disease. Gene therapy using FGF-2 has shown potential in promoting angiogenesis and improving blood flow in ischemic tissues[29,30].HGF gene therapy is designed to stimulate angiogenesis and has been eval-uated in clinical trials for its potential to treat ischemic heart disease and peripheral artery disease[31,32].
Angiopoietin-1 (Ang-1) therapy stabilizes newly formed blood vessels and promotes vascular maturation. It has been studied for its potential to enhance angiogenesis in is-chemic tissues and improve tissue repair[33]. Platelet-derived growth factor-B (PDGF-B) therapy involving PDGF-B has been explored for its role in recruiting pericytes and smooth muscle cells, stabilizing new blood vessels, and improving blood flow in ischemic tissues[34,35]. Hypoxia-Inducible Factor-1 Alpha (HIF-1α) Gene Therapy involves the de-livery of HIF-1α, a transcription factor that induces the expression of several angiogenic factors, including VEGF, under hypoxic conditions. It has been studied for treating is-chemic cardiovascular diseases. Some therapies synergize VEGF and FGF genes to pro-mote angiogenesis and enhance therapeutic outcomes[36,37]. Combining these genes en-courages the formation and stabilization of new blood vessels, providing a more robust angiogenic response[38,39].
Pro-angiogenic peptide drugs are designed to promote the formation of new blood vessels and have potential applications in treating various conditions such as ischemic diseases, wound healing, and tissue regeneration. Thymosin Beta-4 is a small, 43-amino acid peptide. TB-4 promotes angiogenesis by enhancing endothelial cell migration and differentiation. It also regulates actin polymerization and promotes wound healing[40]. VEGF Mimetic Peptidesare designed to mimic the active site of VEGF and typically consist of short sequences derived from the VEGF protein. VEGF mimetic peptides bind to VEGF receptors, activating them to stimulate angiogenesis and endothelial cell proliferation[41]. Angiopoietin-Derived PeptidesThese peptides are derived from angiopoietins, particularly the receptor-binding regions of Ang-1 or Ang-2. They mimic the action of angiopoietins, promoting blood vessel maturation and stability. Ang-1 derived peptides are especially noted for enhancing vascular stabilization[42]. Hepatocyte Growth Factor (HGF) Mimetic PeptidesThese are short peptides derived from the active regions of HGF. HGF mimetic peptides activate the c-Met receptor, promoting angiogenesis and enhancing tissue repair and regeneration[43]. Fibroblast Growth Factor (FGF) Derived PeptidesThese peptides are derived from FGF, particularly the regions that interact with FGF receptors. FGF-derived peptides stimulate endothelial cell proliferation and differentiation, promoting angiogen-esis and tissue repair[44]. R-spondin Peptidesare a family of secreted proteins, and pep-tides derived from them are designed to activate the Wnt signaling pathway. R-spondin peptides promote angiogenesis through the activation of Wnt signaling, which is involved in endothelial cell proliferation and migration[45].
Pro-angiogenic organic molecules, often small molecules, are designed to promote angiogenesis through various mechanisms. Thalidomide has a glutarimide ring attached to a phthalimide ring. Initially known for its teratogenic effects, thalidomide has been found to promote angiogenesis under certain conditions by increasing the expression of VEGF and other pro-angiogenic factors[46]. Vandetanib is a quinazoline derivative. Vandetanib is a tyrosine kinase inhibitor that targets VEGFR, EGFR, and RET kinase, promoting angiogenesis by upregulating VEGF signaling pathways[47]. Sorafenib is a biaryl urea. sorafenib inhibits multiple kinases involved in angiogenesis, including VEGFR and PDGFR. This inhibition can paradoxically lead to pro-angiogenic effects in certain contexts, such as by promoting a more normalized vascular environment[48]. Le-nalidomide is a derivative of thalidomide with an isoindolinone structure.Lenalidomide enhances angiogenesis by increasing VEGF production and other growth factors, similar to thalidomide but with improved safety and efficacy profiles[49]. Bevacizumab is a mon-oclonal antibodyAlthough primarily an anti-angiogenic agent targeting VEGF-A, in cer-tain dosages and contexts, it can paradoxically promote angiogenesis by modifying VEGF signaling and vascular normalization[50]. 2-Methoxyestradiol (2-ME2) is an endogenous estrogen metabolite2-ME2 promotes angiogenesis by stabilizing HIF-1α and upregulating VEGF. It also modulates microtubule dynamics[51].
- Ferrara, N. "Vascular Endothelial Growth Factor: Basic Science and Clinical Progress." Endocrine Reviews, vol. 25, no. 4, 2004, pp. 581–611.
- Nugent, M. A., and Iozzo, R. V. "Fibroblast Growth Factor-2." The International Journal of Biochemistry & Cell Biology, vol. 32, no. 2, 2000, pp. 115–120.
- Fiedler, U., et al. "The Angiopoietin-Tie System in Vascular Development and Disease." Journal of Thrombosis and Haemo-stasis, vol. 1, no. 8, 2003, pp. 1683–1690.
- Heldin, C. H., and Westermark, B. "Mechanism of Action and In Vivo Role of Platelet-Derived Growth Factor." Physiologi-cal Reviews, vol. 79, no. 4, 1999, pp. 1283–1316.
- Hinck, A. P., Mueller, T. D., and Springer, T. A. "Structural Biology and Evolution of the TGF-β Family." Cold Spring Harbor Perspectives in Biology, vol. 8, no. 12, 2016, a022103.
- Carpenter, G., and Cohen, S. "Epidermal Growth Factor." Annual Review of Biochemistry, vol. 48, no. 1, 1979, pp. 193–216.
- Nakamura, T., and Mizuno, S. "The Discovery of Hepatocyte Growth Factor (HGF) and Its Significance for Cell Biology, Life Sciences and Clinical Medicine." Proceedings of the Japan Academy, Series B, Physical and Biological Sciences, vol. 86, no. 6, 2010, pp. 588–610.
- Visse, R., and Nagase, H. "Matrix Metalloproteinases and Tissue Inhibitors of Metalloproteinases: Structure, Function, and Biochemistry." Circulation Research, vol. 92, no. 8, 2003, pp. 827–839.
- Carmeliet, P. "VEGF as a Key Mediator of Angiogenesis in Cancer." Oncology, vol. 69, suppl. 3, 2005, pp. 4-10.
- Simons, M., et al. "Clinical Trials in Coronary Angiogenesis: Issues, Problems, Consensus: An Expert Panel Summary." Cir-culation, vol. 102, no. 11, 2000, pp. e73-e86
- Symes, J. F., et al. "Gene Therapy with Acidic Fibroblast Growth Factor (FGF-1) for Myocardial Angiogenesis." Annals of Thoracic Surgery, vol. 68, no. 2, 1999, pp. 830-836.
- Giacca, M., and Zacchigna, S. "VEGF Gene Therapy: Therapeutic Angiogenesis in the Clinic and Beyond." Gene Therapy, vol. 19, no. 6, 2012, pp. 622-629.
- Nakamura, T., Mizuno, S. "The Discovery of Hepatocyte Growth Factor (HGF) and Its Significance for Cell Biology, Life Sciences and Clinical Medicine." Proceedings of the Japan Academy, Series B, Physical and Biological Sciences, vol. 86, no. 6, 2010, pp. 588-610.
- Kanda, H., et al. "Clinical Applications of Angiogenic Gene Therapy." Regenerative Therapy, vol. 3, 2016, pp. 37-44.
- Thurston, G., and Daly, C. "The Complex Role of Angiopoietin-2 in the Angiopoietin-Tie Signaling Pathway." Cold Spring Harbor Perspectives in Medicine, vol. 2, no. 9, 2012, a006550.
- Raines, E. W. "PDGF and Cardiovascular Disease." Cytokine & Growth Factor Reviews, vol. 15, no. 4, 2004, pp. 237-254.
- Rutanen, J., et al. "Gene Therapy with AdVEGF-B Promotes Angiogenesis and Induces Recovery of Distal Ischemic Tissues in Diabetic Mice." Cardiovascular Research, vol. 61, no. 1, 2004, pp. 158-168.
- Rajendran, V., et al. "Hypoxia-Inducible Factor (HIF)-1α: A Novel Target for Cardioprotection." Current Pharmaceutical Design, vol. 21, no. 2, 2015, pp. 241-247.
- Harada, K., et al. "Gene Therapy with DNA Encoding Hypoxia-Inducible Factor-1α Improves Angiogenesis and Perfusion in a Rabbit Model of Chronic Myocardial Ischemia." American Journal of Physiology-Heart and Circulatory Physiology, vol. 296, no. 3, 2009, pp. H825-H833.
- Nakanishi, K., et al. "Synergistic Effect of Dual Gene Therapy with Hepatocyte Growth Factor Plus Vascular Endothelial Growth Factor and Low-Energy Shock Wave Therapy for Erectile Dysfunction in a Rat Model." Journal of Sexual Medicine, vol. 7, no. 9, 2010, pp. 3338-3346.
- Gruchala, M., et al. "Gene Therapy with Multiple Vectors and Multifactorial Approaches in Cardiovascular Medicine." Trends in Cardiovascular Medicine, vol. 13, no. 2, 2003, pp. 8-14.
- Jerry R Mendell , Samiah A Al-Zaidy , Louise R Rodino-Klapac , Kimberly Goodspeed , Steven J Gray , Christine N Kay , Sanford L Boye , Shannon E Boye , Lindsey A George , Stephanie Salabarria , Manuela Corti , Barry J Byrne , Jacques P Tremblay . Mol Ther 2021 ;29(2):464-488.
- Goldstein, A. L., and Kleinman, H. K. "Thymosin β4: Actin Sequestering Protein Moonlights to Repair Injured Tissues." Trends in Molecular Medicine, vol. 11, no. 9, 2005, pp. 421-429.
- Fairbrother, W. J., et al. "VEGF mimetics as potent activators of the VEGF receptor and angiogenesis." PNAS, vol. 103, no. 9, 2006, pp. 3226-3231.
- Kim, I., et al. "Angiopoietin-1-derived synthetic peptides and their effects on endothelial cell function and angiogenesis." Journal of Biological Chemistry, vol. 275, no. 3, 2000, pp. 2070-2074.
- Nakamura, T., Mizuno, S. "The discovery of Hepatocyte Growth Factor (HGF) and its significance for cell biology, life sci-ences and clinical medicine." Proceedings of the Japan Academy, Series B, Physical and Biological Sciences, vol. 86, no. 6, 2010, pp. 588-610.
- Presta, M., et al. "Fibroblast growth factor/fibroblast growth factor receptor system in angiogenesis." Cytokine & Growth Factor Reviews, vol. 16, no. 2, 2005, pp. 159-178.
- Kazanskaya, O., et al. "R-spondin2 is a secreted activator of Wnt/β-catenin signaling and is required for Xenopus myogene-sis." Developmental Cell, vol. 7, no. 4, 2004, pp. 525-534.
- D'Amato, R. J., et al. "Thalidomide is an inhibitor of angiogenesis." Proceedings of the National Academy of Sciences, vol. 91, no. 9, 1994, pp. 4082-4085.
- Carlomagno, F., et al. "Vandetanib (ZD6474), a dual inhibitor of RET and VEGFR-2 tyrosine kinases, has potent anti-tumor activity in RET-associated cancers." The Journal of Clinical Endocrinology & Metabolism, vol. 89, no. 12, 2004, pp. 5798-5805.
- Wilhelm, S. M., et al. "Bay 43-9006 exhibits broad spectrum oral antitumor activity and targets the RAF/MEK/ERK pathway and receptor tyrosine kinases involved in tumor progression and angiogenesis." Cancer Research, vol. 64, no. 19, 2004, pp. 7099-7109.
- Zhu, Y. X., et al. "Lenalidomide enhances T cell and N.K. cell-mediated cytotoxicity and decreases tumor growth in a murine myeloma model." Leukemia, vol. 22, no. 4, 2008, pp. 665-672.
- Hurwitz, H., et al. "Bevacizumab plus irinotecan, fluorouracil, and leucovorin for metastatic colorectal cancer." New Eng-land Journal of Medicine, vol. 350, no. 23, 2004, pp. 2335-2342.
- Attalla, H., et al. "2-Methoxyestradiol induces apoptosis and inhibits vascular endothelial growth factor-induced angiogene-sis in human breast cancer cells." Cancer Research, vol. 56, no. 24, 1996, pp. 5590-5594.
Comment 2: 2-The whole manuscript should be reviewed for correction editing and typing mistakes like:
Response 2: the manuscript was revised for errors and typos.
Comment 3 : Line 118: Correct --12.71 to -12.71
Response 3: Corrected
Comment 4 : Put the reference No. 13 before the dot not after.
Response 4: Corrected
Comment 5 : Line 493: The references [35,36,37] corrected to [35-73]
Response 5: Corrected as suggested.
Comment 6: The title of table 1 should be placed above the table not below it.
Response 6: Corrected as suggested
Comment 7: 3- Write a short paragraph before the conclusion discussing the future prospects of this study
Response 7: the fpoollowing thext was added as suggested: This study provides an overview of the main features of molecules that either inhibit or stimulate angiogenesis. The findings of the study can be utilized to create a potent stimulator of angiogenesis. Both wet lab experiments and computational methods are necessary to accomplish this objective. The resulting molecular systems can then be employed to develop a pharmacologically active stimulator of angiogenesis, which can target either inhibitors or multiple targets simultaneously.
Comment 8: 4- References are written with a wrong format. The year should be in bold without bracket and the name of the journal should be italic.
Response 8: the references have been corrected.
Comment 9: 5-The author said there is a supplementary file S1 while there was no attached supplementary file and the link for supplementary materials didn't work.
Response 9: corrected as suggested
Comment 10: The English language is fine except some few mistakes need correction
Response 10: minor typos have been corrected
Reviewer 2 Report
Comments and Suggestions for Authors
In this work, the authors employed computational docking and molecular modelling to analyse how different molecules known to stimulate or inhibit angiogenesis interact with the key VEGFR2 protein involved in the process. The authors report that the chemical spaces of stimulators and inhibitors are similar, with a main difference that inhibitors occupy a more expanded and diverse chemical space. This conclusion is derived from the fact that the inhibitory molecules are appearing quite different from one another, and they showed high energetical binding to the VEGFR2.
The design and workflow of this work are generally good. However, there are some important aspects that I would like to see improved prior to considering it for publication.
a) There are limitations to docking studies, such as inaccurate scoring functions, inadequate treatment of protein flexibility and solvent effects, and limited conformational sampling, which can lead to incorrect predictions. The docking studies often provide a static snapshot and overlook the dynamic, complex nature of biological interactions. These limitations can impact the reliability of the results. The authors could have placed a paragraph in the discussion/result section discussing these aspects and how they potentially influence the interpretations of their work.
b) The authors obviously did docking studies leading to calculated structures and geometries. These geometries are part of their produced data and they need to be made open. Statement such as “Data Availability Statement: Not applicable” is not appropriate in this case.
c) The article’s structure would be more logical to start with introduction, methods, results and discussion and then conclusion. Currently the methods section comes very late which is difficult to follow.
d) Methods section: This section needs to contain the most essential information, which is currently a single paragraph. The 27 compounds and their smile strings can be shown as drawings, but their place is ideally in a supporting information document. Next. The authors go to a great extent in the methodology section to explain how they have derived every molecule. That is good. Instead of writing a very long text, they could have a separate table as part of the supporting information. This can be a table with “compound”, “motivated by” – giving the reason why, and the relevant “reference.”
e) I suggest merging the results and the discussion section. The authors can provide a result, which may be a figure, present it as it is and then provide their discussion. The discussion should be also limited to the work, currently it extends very long and does not directly reports on the figures.
f) Some suggestions regarding the formatting of the Results and Discussion section:
Figure 1: No need to write Models (Model 01, 02, 03) in the same line as it is clear that all of these are models.
Figure 2. This figure has multiple subfigures. If the authors show the cavity then it should be shown a subsection where that cavity originates from.
Figure 2. Please do not list smiles; use the compound name instead.
Table 1: All units need to be in the names of the columns.
Tables 1,2 and 3 are basically energies. Please represent them with the help of XY diagrams and keep all data as part of the supporting information.
Table 6 – please shift in the supporting information. Ideally please redraw all structures and place them as part of the methodology. You can keep labeling them as compound 1 ,2 ….. 27 and refer to them in the main text.
g) The so called chemical space – on Figures 6 and 7. These are typical cheminformatics parameters that are calculated. I am aware that they can be automatically done with RDkit. However the current representation is not sensical. It could have been more accurate to present as a single table. Currently you indicate a molecular weight of 40000+. Is the scale also for Number of H bonds? I think it is not really clear and accurate. Please adjust.
h) References: please add doi to all papers.
Comments on the Quality of English LanguageIn the affiliation line, the authors write: "Faculty of Medicine and Pharamacy". It should be "Faculty of Medicine and Pharmacy". The word Pharmacy is incorrectly written. Also, for consistency, the authors should change "homology I.D." to "homology ID".
Author Response
Reviewer 2
In this work, the authors employed computational docking and molecular modelling to analyse how different molecules known to stimulate or inhibit angiogenesis interact with the key VEGFR2 protein involved in the process. The authors report that the chemical spaces of stimulators and inhibitors are similar, with a main difference that inhibitors occupy a more expanded and diverse chemical space. This conclusion is derived from the fact that the inhibitory molecules are appearing quite different from one another, and they showed high energetical binding to the VEGFR2.
The design and workflow of this work are generally good. However, there are some important aspects that I would like to see improved prior to considering it for publication.
Comment 1: a) There are limitations to docking studies, such as inaccurate scoring functions, inadequate treatment of protein flexibility and solvent effects, and limited conformational sampling, which can lead to incorrect predictions. The docking studies often provide a static snapshot and overlook the dynamic, complex nature of biological interactions. These limitations can impact the reliability of the results. The authors could have placed a paragraph in the discussion/result section discussing these aspects and how they potentially influence the interpretations of their work.
Response 1: Corrected as suggested. The folowing text and references were added: Docking studies have certain drawbacks, including imprecise scoring functions, insufficient consideration of protein flexibility and solvent effects, and restricted conformational sampling, all of which can result in inaccurate predictions. Docking studies frequently prefer a fixed image and fail to consider the dynamic and intricate characteristics of biological interactions. These constraints can affect the dependability of the outcomes[26,27].
However, the utilization of docking in drug design is restricted to biological targets that have known crystal structures. Various methods have been employed to address this specific constraint. One way to overcome the lack of 3D structures is to create homology models using structural templates that have very similar sequences. In addition, these techniques can be employed in conjunction with molecular dynamics (M.D.) to corroborate and enhance the accuracy of the computationally simulated complexes [28,29]. However, the current advancements in structural biology and crystal structure determination, which are steadily improving the availability of experimentally obtained ligand-target complexes [36,37,38,39], will undoubtedly alleviate this problem. Computational techniques, such as molecular dynamics, have been extensively employed to investigate the conformational space of the targets, ligands, and ligand-target complexes. This allows for a more accurate description of the dynamic behavior of ligand-target complexes and improves the precision of docking results[30,31].In this respect, the computational studies performed in this work have been performed using crystallographic models with the best resolution possible. Also, the protein-protein complexes were selected based on the most favorable complex energies. Furthermore, each complex was subject again to energy minimization and structural error detection methodologies
- Elokely K.M., Doerksen R.J. Docking challenge: Protein sampling and molecular docking performance. J. Chem. Inf. Model. 2013;53:1934–1945
- Pantsar T., Poso A. Binding affinity via docking: fact and fiction. Molecules. 2018;23:1899.
- Salmaso V., Moro S. Bridging molecular docking to molecular dynamics in exploring ligand-protein recognition process: An overview. Front. Pharmacol. 2018;9:923
- De Vivo M., Masetti M., Bottegoni G., Cavalli A. Role of molecular dynamics and related methods in drug discovery. J. Med. Chem. 2016;59:4035–4061.
- Durrant J.D., McCammon J.A. Molecular dynamics simulations and drug discovery. BMC Biol. 2011;9:71.
- Karplus M., McCammon J.A. Molecular dynamics simulations of biomolecules. Nat. Struct. Biol. 2002;9:646–652.
Comment 2 b) The authors obviously did docking studies leading to calculated structures and geometries. These geometries are part of their produced data and they need to be made open. Statement such as "Data Availability Statement: Not applicable" is not appropriate in this case.
Response 2 : Corrected as suggested: The complexes have been made available supplemental material S1. Also the statement was corrected: Data Availability Statement: Available on reasonable demand.
Comment 3c) The article's structure would be more logical to start with introduction, methods, results and discussion and then conclusion. Currently the methods section comes very late which is difficult to follow.
Response 3: The suggestion of the respected reviewer is reasonable; however, this is the current format of ijms provided as a template.
Comment 4 d) Methods section: This section needs to contain the most essential information, which is currently a single paragraph. The 27 compounds and their smile strings can be shown as drawings, but their place is ideally in a supporting information document. Next. The authors go to a great extent in the methodology section to explain how they have derived every molecule. That is good. Instead of writing a very long text, they could have a separate table as part of the supporting information. This can be a table with "compound", "motivated by" – giving the reason why, and the relevant "reference."
Response 4: The section is composed of multiple paragraphs. Table 6 was removed and transferred to the supplemental material S2, as suggested. Regarding the suggestion of writing separate tables supported by references, we prefer the narrative and a more comprehensive type of presentation of the material and methods.
Comment 5 e) I suggest merging the results and the discussion section. The authors can provide a result, which may be a figure, present it as it is, and then offer their discussion. The discussion should also be limited to the work; currently, it has been extended a very long time and does not directly report on the figures.
Response 5: Most of the results deal with biochemistry or mathematical-mathematical chemistry abstract notions. The results are discussed point by point in the discussion section in the order they appear in the text. Each figure, drawing, and computation presented is discussed in specific detail. The discussion of each result starts from a general point of view and goes into specific details and actual interpretation of the result. For instance results in table 2 and 3 are dicusses starting with g the general aspects, characterization and dicussion of all the terms and finally getting intop specific details ………….. When the bond angles deviate, the system's potential energy increases, contributing to the overall energy of the molecule. Here, both angular and solubility energies show favorable values that correlate with the total complex energies. Overall, docking results show that the docking procedure was performed correctly. Finally, VEGFR2 forms stable active com-plexes with the inhibitory and stimulant peptides retrieved from the literature[24,25,26]. However, all complexes of the of the inhibitory and stimulatory proteins display favorable energies with presumably notable biological activity.Regarding inhibitory molecules docking energies, the most favorable energy is observed at 4EB1 with a total complex en-ergy of -92.87 kcal/mol. The highest docking energy observed at stimulants molecule is observed at 2X1W with a docking energy of -99.99 kcal/mol. Also, in the case of inhibitors, the most favorable solvation energy is observed at 4EB1 with 15734.68 kcal/mol. The same is true in the case of the stimulants; the most favorable docking energy is observed at 2XIW with -14554.78 kcal/mol….
-this path is followed for the whole results in order to make the results clear and understandable for all readers of the manuscript
-The journal format includes both results sections, where raw results must be presented, and a discussion section, where discussions must be carried out based on the results.
-in this case, given the relative complexity of the manuscript, results are presented as simply as possible and discussed in detail in the discussion section in order of their appearance in the manuscript. This increases the accessibility and understandability of the manuscript for the general reader.
Some suggestions regarding the formatting of the Results and Discussion section:
Comment 6 Figure 1: No need to write Models (Model 01, 02, 03) in the same line as it is clear that all of these are models.
Response 6: corrected as suggested. The word model was deleted.
Comment 7 Figure 2. This figure has multiple subfigures. If the authors show the cavity, then a subsection should be shown where that cavity originates.
Response 7: The figure was redrawn according to suggestion. Also, the following text was added at the legend- including the coordinates of the binding site: the coordinates of the binding site are shown by black arrow (x=29.04Å;y=-36.68Å;z=-18.54Å), a detail of the binding site is also represented together with a space-filling of the binding site.
Comment 8: Figure 2. Please do not list smiles; use the compound name instead.
Response 8: smiles converted to name as suggested
Comment 9:- Table 1: All units need to be in the names of the columns.
Response 9: Numbers were converted to names as suggested. Also, as indicated in the previous comments, the list was submitted as a supplementary file.
Comment 10: Tables 1,2 and 3 are basically energies. Please represent them with the help of X.Y. diagrams and keep all data as part of the supporting information.
Response 10: corrected as suggested. Table 1,2,3 has been converted to scatter plots as suggested
Comment 11: Table 6 – please shift in the supporting information. Ideally please redraw all structures and place them as part of the methodology. You can keep labeling them as compound 1 ,2 ….. 27 and refer to them in the main text.
Response 11: As suggested previously, Table 6 was converted into supplemental material.
Comment 12 g) The so called chemical space – on Figures 6 and 7. These are typical cheminformatics parameters that are calculated. I am aware that they can be automatically done with RDkit. However the current representation is not sensical. It could have been more accurate to present as a single table. Currently you indicate a molecular weight of 40000+. Is the scale also for Number of H bonds? I think it is not really clear and accurate. Please adjust.
Response 12: The computations are accurate. For the reviewer's convenience, we have attached the supplemental file S4. Note that there is a difference between the molar mass and the molecular weight, which is computed here as a descriptor. In mathematical chemistry, the molecular weight is used more often. At the same time, it can serve as a molecular descriptor ( it has a higher value than the molecular mass and gives specific( un-degenerate values). Here is a brief discussion of Molar mass vs. molecular weight:
It may seem as though the two quantities mean the same thing, but this is not true.
Molecular weight (or molecular mass) is the mass of a molecule given in the Dalton unit (Da) or the unified atomic mass unit (u). This one roughly corresponds to the mass of a single proton or neutron. For example, the molecular weight of CO2 is 44.01 Da or 44.01 u.- so in a protein, many protons explain the 150.000 value observed by the reviewer.
Morvore, the chemical spaces are similar – in shape- for the free molecules and the chimeric models. So, there is no error.
Radial graphs are a customary representation of such computations. We prefer to keep this representation while it is easier to read and understand instead of converting it into a table. Also, as stated before, the raw data, together with the graph, were organized in an Excel table and submitted as supplemental material S4.
Comment 13 h) References: please add doi to all papers.
Response 13 : It is customary not to use doi in the reference. The ijms template does not allow the addition of doi to a reference. From past and present experiences in publishing with MDPI the doi had to be removed from all references.
Comment 14 In the affiliation line, the authors write: "Faculty of Medicine and Pharamacy". It should be "Faculty of Medicine and Pharmacy". The word Pharmacy is incorrectly written. Also, for consistency, the authors should change "homology I.D." to "homology I.D.".
Response 14: corrected as suggested.
Round 2
Reviewer 2 Report
Comments and Suggestions for Authors
The authors have positively answered my concerns.
The data file is comprehensive, and it needs to be made public
Thus, do not write "Data Availability Statement: Available on reasonable demand."
but provide the data as direct supporting information to this paper. Alternatively, the data can be made public via permanent doi from the university library or publishing it via Zenodo or Figshare. If the authors publish it via other repositories with permanent links, they can do so.
Prior to publication, the authors need to update the references with doi links.
Author Response
Comment1: The authors have positively answered my concerns. The data file is comprehensive, and it needs to be made public Thus, do not write "Data Availability Statement: Available on reasonable but provide the data as direct supporting information to this paper. Alternatively, the data can be made public via permanent doi from the university library or publishing it via Zenodo or Figshare. If the authors demand." publish it via other repositories with permanent links, they can do so. Prior to publication, the authors need to update the references with doi links.
Response 1: The manuscript has been corrected as suggested. The data available statements were deleted and replaced with a phrase indicating s1,s2,s3,s, and 4.
Thank you for reviewing our manuscript.